# Diversity in Large Language Models under Supervised Fine-Tuning

## Abstract

Supervised fine-tuning (SFT) is essential for aligning Large Language Models (LLMs) with user intent, yet it is frequently observed to suppress generative diversity. Although this reduction is often discussed, formal testing remains limited and rarely covers a large scale. In this study, we conduct an extensive evaluation that confirms this decline and attribute it to two primary drivers: the neglect of low-frequency patterns within fine-tuning datasets and the forgetting of preexisting knowledge. Although certain SFT loss functions attempt to address these challenges, they typically focus on a single issue in isolation, and some promising objectives remain underutilized for LLM fine-tuning. We perform a rigorous analysis of these potential candidates to identify their most effective components. By synthesizing these theoretically grounded elements, we develop a novel objective that simultaneously preserves pretraining knowledge and ensures a balanced treatment of underrepresented SFT samples. Our rigorous evaluation reveals key insights into the reduction in model expressiveness during alignment. Across multiple models and benchmarks, we demonstrate that our proposed objective enhances output diversity while preserving high response quality, offering a principled approach to LLM fine-tuning.

## 1. Introduction

Autoregressive language models (LMs) (Bengio et al., 2000) have demonstrated remarkable progress in modeling natural language. Increased data availability and model capacity have allowed Transformer-based (Vaswani et al., 2023) architectures to generate text that closely resembles human-written content. Modern Large Language Models (LLMs) now serve as powerful generative engines that excel across

a broad range of specialized tasks, from document summarization (Brown et al., 2020) to complex reasoning (Wei et al., 2023).

Despite their impressive capabilities, pretrained LLMs often produce responses that do not fully capture users' intents. This limitation arises because, even with extensive pretraining, models may generate tokens that are statistically plausible yet semantically misaligned with the actual query, resulting in irrelevant or unhelpful outputs. To address this issue, researchers increasingly rely on instruction tuning (Raffel et al., 2023; Wei et al., 2022; Chung et al., 2022), also referred to as Supervised Fine-Tuning (SFT) (Ouyang et al., 2022; Bai et al., 2022). This process refines LLMs on curated collections of high-quality prompt–response pairs, enabling better alignment with user expectations and task-specific objectives. The SFT stage plays a crucial role in preparing models for subsequent reinforcement learning (RL) alignment. By first grounding the model in high-quality demonstrations, SFT provides a stable initialization that enables RL methods to effectively refine task-specific behaviors, rather than struggling with unstructured or misaligned model outputs.

However, SFT also introduces notable challenges. While pretrained LLMs naturally generate diverse outputs, fine-tuning is considered to reduce this diversity, as has been observed for specific model families and evaluation sets (O'Mahony et al., 2024). To mitigate these effects, prior work has explored various regularization strategies, including weight decay during fine-tuning (Krogh & Hertz, 1991), noise injection into input embeddings (Jain et al., 2023), model ensembling (Hao et al., 2025), and alternative loss formulations (Li et al., 2024; Verine et al., 2025). Although these methods offer various improvements, they are inspired by different analytical paradigms and do not fully resolve the underlying factors limiting generative diversity, leaving room for further refinement.

In this work, we attribute the reduction in output diversity during SFT to the interplay of two factors (Figure 1). First, fine-tuning on small, curated datasets often induces overfitting to dominant patterns (Bethune et al., 2025), causing the model to neglect the long-tail distribution of the data. Second, the process can erode the diverse knowledge acquired during pretraining (Goodfellow et al., 2015), further narrow-

[1]Anonymous Institution, Anonymous City, Anonymous Region, Anonymous Country. Correspondence to: Anonymous Author <anon.email@domain.com>.

Preliminary work. Under review by the International Conference on Machine Learning (ICML). Do not distribute.

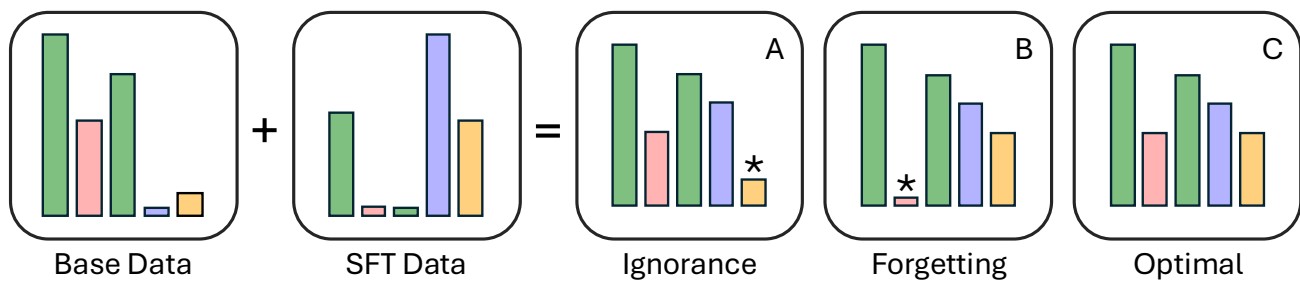

*Figure 1.* Impact of SFT on Generative Diversity. Comparison of data distributions and model states: Base Data represents the broad pretraining corpus, SFT Data represents the curated instruction set. (A) **Ignorance** illustrates the failure to capture low-frequency SFT patterns, while (B) **Forgetting** depicts the erosion of the original pretraining knowledge. (C) **Optimal** represents the ideal balance where the model integrates new instructions without sacrificing the generative flexibility or richness of the base distribution.

ing the model's available response space. The combination of these two phenomena leads to an overall diminished diversity.

To counter these issues, we focus on loss-based regularization as a principled solution. We investigate existing SFT functions to determine their ability to preserve pretrained knowledge. In addition, we also evaluate Focal Loss (FL) (Lin et al., 2017) as a potential SFT objective for maintaining the balanced treatment of rare samples. Building on these insights, we propose a new Tempered Focal (TOFU) loss function explicitly designed to tackle both forgetting and ignorance simultaneously. Our main contributions can be summarized as follows:

- We conduct an extensive evaluation of existing SFT objectives across various model families, datasets, and benchmarks.

- We investigate Focal Loss as an SFT objective for maintaining balanced treatment of rare samples and demonstrate its promising performance.

- We propose a novel training objective, TOFU, designed to mitigate both ignorance and forgetting.

- We show that our method achieves superior generative diversity while maintaining competitive output quality compared to standard approaches.

## 2. Theoretical preliminaries and backgrounds

**Large Language Models** Large Language Models are trained as next-token predictors over a discrete vocabulary $\mathcal{V}$. Given a sequence of tokens $x_{1:L} = (x_1, ..., x_L)$, the model defines a conditional distribution over the next token $x_l$ for each vocabulary element $y \in \mathcal{V}$:

$$p_\theta(y|x_{<l}) \doteq p_\theta(x_l = y|x_{<l}). \qquad (1)$$

Training consists of minimizing the Cross-Entropy (CE) between the model's predicted distribution and a target distribution $q \doteq q(y|x_{<l})$ at each position:

$$\mathcal{L}_{CE}(\theta) = -\sum_{l=1}^{L} \mathbb{E}_q \log p_\theta(y|x_{<l}) \qquad (2)$$

For standard supervised training, $q$ is one-hot on the ground-truth token, making the training equivalent to maximizing the likelihood of the observed sequence. For simplicity, throughout this paper we focus on the loss corresponding to a single token, without explicitly showing its dependence on the context, as this omission is purely notational and does not change the underlying mathematics.

**Supervised Fine-Tuning** In Supervised Fine-Tuning, LLMs are adapted to specific tasks using sequences that combine a prompt and a response. The model conditions on the prompt as a fixed prefix but is optimized exclusively on the response. This ensures focus on generating the correct outputs for the task, while leveraging existing pretrained knowledge.

The standard training objective for SFT is the Cross-Entropy loss. It has a significant limitation: on relatively small fine-tuning datasets, it encourages the model to focus narrowly on the few observed responses while ignoring other plausible outputs (Li et al., 2024). Therefore CE can reduce the expressiveness of the model's generation, a property that is important for downstream exploration and robust alignment. To address this issue, recent work in the community has explored variants of SFT that employ modified loss functions specifically designed to preserve or enhance output diversity.

**GEM by Li et al. (2024)** GEM (Game-theoretic Entropy Maximization) reframes supervised fine-tuning as a distribution-matching process in which learning is modeled as transferring probability mass from non-target to target

tokens. Instead of relying on Cross-Entropy, which forces indiscriminate and unbounded probability flow, GEM introduces a game-theoretic formulation. In this setup, a meta-controller regulates how and where probability mass moves. This selective control prevents collapse of the output distribution and avoids over-penalizing semantically meaningful or rare tokens. The game-theoretic formulation ultimately yields a practical training algorithm in which the entire framework reduces to optimizing a new Cross-Entropy replacing loss function.

**Definition 2.1** (GEM loss (Li et al., 2024))**.** The GEM's objective is defined as follows:

$$\mathcal{L}_{GEM}(\theta) = -\mathbb{E}_q[\log p_\theta] + \mathbb{E}_{p_\theta^\beta}[\log p_\theta], \qquad (3)$$

where $p_\theta^\beta \doteq \text{softmax}\left(\beta^{-1} \log p_\theta\right)$ is a temperature-scaled distribution with $\beta \in (0, 1)$ as a temperature parameter. Note that $p_\theta^\beta$ is detached from gradients computation. Here and throughout this work we highlight detached gradients in red.

Taken together, GEM produces sparse, targeted updates that preserve useful pretraining knowledge while still aligning the model to the supervised dataset. The reduced forgetting results in greater output diversity, as the model maintains a broader and more balanced token distribution.

**$\lambda$-PR by Verine et al. (2025)** $\lambda$-PR is a training objective for language models that explicitly manages the trade-off between quality and diversity, building on previous approaches that improve generation. It draws on ideas such as removing high-loss examples to reduce the impact of noisy references (Kang & Hashimoto, 2020), downweighting unlikely sequences to prevent degeneration (Ji et al., 2023), and reweighting gradients to promote high-probability outputs (Pang & He, 2021). Although these techniques were originally intended to enhance quality, $\lambda$-PR partially reverses their effects to optimize for diversity. The resulting compound loss function provides a unified framework for fine-tuning the balance between the two competing metrics.

**Definition 2.2** ($\lambda$-PR loss (Verine et al., 2025))**.** The $\lambda$-PR's objective is defined as follows:

$$\mathcal{L}_{\lambda-\text{PR}}(\theta) = -\mathbb{E}_q\left[w(\lambda, \alpha) \log p_\theta\right], \qquad (4)$$

where $w(\lambda, \alpha) = \lambda^{\frac{l-1}{L}} \mathbb{I}_{p_\theta \leq \delta} \frac{p_\theta}{\alpha + (1-\alpha)p_\theta}$. Here, $\lambda \in \mathbb{R}^+$ is the main parameter, which controls a trade-off, $\alpha \in [0, 1]$ and $\delta = \frac{\alpha\lambda^{1/L}}{1-(1-\alpha)\lambda^{1/L}}$ ($L$ being sequence length and $l$ token's position).

$\lambda$-PR can be interpreted as a weighted Cross-Entropy loss, where different tokens contribute unequally to the overall objective. The loss is motivated by two complementary goals: removing high log-loss examples to encourage a broader range of outputs, and downweighting less probable tokens to preserve accuracy.

**Focal Loss by Lin et al. (2017)** Focal Loss, previously widely used in computer vision for unbalanced training (Lin et al., 2017), has been relatively overlooked in the context of large language models. It addresses class imbalance by reshaping the standard Cross-Entropy loss to downweight well-classified examples.

**Definition 2.3** (Focal Loss (Lin et al., 2017))**.** The Focal Loss is defined as follows:

$$\mathcal{L}_{FL}(\theta) = -\mathbb{E}_q\left[(1 - p_\theta)^\gamma \log p_\theta\right], \qquad (5)$$

where power coefficient $\gamma \geq 0$ is typically selected from range $[1, 5]$ (Lin et al., 2017; Mukhoti et al., 2020; Charoenphakdee et al., 2020).

We hypothesize that this loss can enhance the model's diversity by placing greater emphasis on underrepresented examples in the SFT dataset.

## 3. Proposed Method

Forgetting and ignorance arise from different aspects of the training dynamics. We propose to address both by combining the forgetting mitigation of GEM with the weighting of Focal Loss for underrepresented samples. However, combining the objectives is not straightforward, as adding terms or factors can significantly alter the update structure, leading to unstable optimization or poor interpretability. This motivates a careful analysis at the gradient level.

By examining GEM's gradients, we gain both theoretical insight and practical benefits. Specifically, our analysis reveals that GEM is equivalent to a temperature-scaled Cross-Entropy loss (Theorem 3.1).

**Theorem 3.1** (GEM loss equivalence)**.** *Training with $\mathcal{L}_{GEM}$ is equivalent to training with temperature-scaled Cross-Entropy loss, as*

$$\nabla_\theta \mathcal{L}_{GEM}(\theta) = \nabla_\theta \mathcal{L}_{CE}^\beta(\theta), \qquad (6)$$

*where $\mathcal{L}_{CE}^\beta(\theta) \doteq -\beta\mathbb{E}_q \log p_\theta^\beta$.*

The equivalence clarifies GEM's advantage over methods that explicitly relax the predictive distribution by adding an entropy term. Instead of uniform flattening, GEM exerts softer pressure via adaptation, preventing the model from becoming overly confident while still guiding it toward the target. For practitioners, this equivalence also simplifies the computation required to implement GEM.

Theoretically, the GEM objective shares a global minimum with Cross-Entropy when the latter is scaled by $1/\beta$ after

fine-tuning. In practice, however, the model does not converge to that point due to the limited updates in SFT and inherent optimization noise. This allows GEM to reach a parameter solution distinct from that of standard post-hoc temperature scaling (Ficler & Goldberg, 2017).

We next analyze Focal Loss in a similar manner. Prior work (Mukhoti et al., 2020) has shown that despite not being a proper loss it can be interpreted as a gradient-scaled Cross-Entropy. For completeness, we restate this result and explicitly state the assumptions required for the equivalence.

**Proposition 3.2** (Focal Loss equivalence for one-hot targets). *Assume that the target distribution $q$ is one-hot. Then, for Focal Loss $\mathcal{L}_{FL}$ and Cross-Entropy $\mathcal{L}_{CE}$, the gradients satisfy*

$$\nabla_\theta \mathcal{L}_{FL}(\theta) = g(\hat{p}_\theta, \gamma)\nabla_\theta \mathcal{L}_{CE}(\theta), \qquad (7)$$

*where $g(p, \gamma) = (1-p)^\gamma - \gamma p(1-p)^{\gamma-1}\log p$. Here and throughout this work $\hat{p}$ denotes the predicted probability assigned to the ground-truth token.*

The scaling function $g(p, \gamma)$ increases the weight of gradients corresponding to underrepresented samples, while reaching its maximum at a probability $p \neq 0$ (Figure 2). This behavior echoes the idea of Trunc loss (Kang & Hashimoto, 2020), which removes the hardest examples, specifically those with predicted probability below a small threshold, from training. In Focal Loss, the fact that the gradient magnitude peaks at a small but nonzero $p$ similarly reduces emphasis on extremely hard examples, while still focusing learning on moderately difficult samples.

Comparing FL scaling with that of the $\lambda$-PR (Figure 2), one might notice the latter's downside: it cannot account for a previously zeroed-out probability $p$ without assigning it the highest possible weight. Consequently, the $\gamma$ parameter of FL provides more flexibility in steering the weighting balance than the $\delta$ of $\lambda$-PR.

The equivalences established by Theorem 3.1 and Proposition 3.2 allow us to merge the core ideas of GEM and Focal Loss by applying the focal term to the temperature-scaled Cross-Entropy.

**Corollary 3.3.** *If $q$ is one-hot, then weighting $\mathcal{L}_{CE}^\beta(\theta)$ by the focal term $(1 - p_\theta^\beta)^\gamma$ results in a gradient of the form $g(\hat{p}_\theta^\beta, \gamma)\nabla_\theta \mathcal{L}_{CE}^\beta(\theta)$.*

However, as follows from Corollary 3.3, a naive multiplication of $\mathcal{L}_{CE}^\beta(\theta)$ by the focal term can excessively amplify the influence of very small probabilities in gradient weighting. This occurs because temperature-scaling reduces these probabilities further, which in turn increases the corresponding focal weight, potentially leading to unintended overemphasis on hard examples. Ideally, one would like to scale the GEM gradients using the focal scaling function $g(p, \gamma)$ evaluated on the unscaled probabilities $p$. Simply

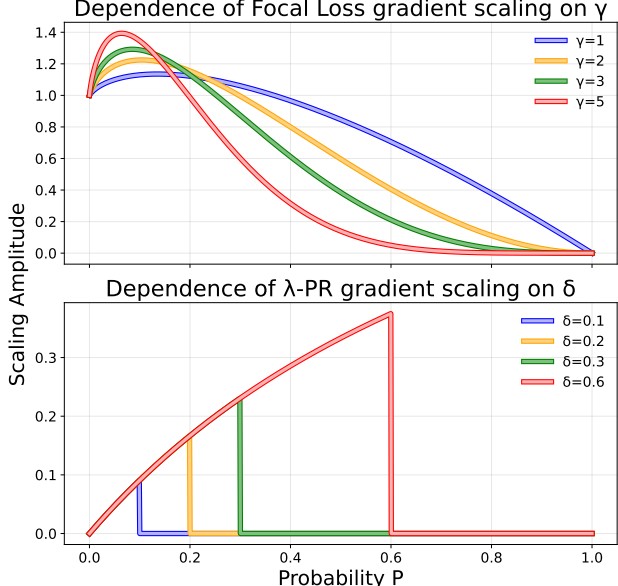

*Figure 2.* Gradient scaling amplitudes for Focal Loss and $\lambda$-PR as functions of various parameters and probabilities $p$. For $\lambda$-PR we omit the dependence on tokens position for simplicity. We also set $\alpha$ to the default value of $10^{-5}$ and do not investigate its impact further due to negligible influence on the function's shape.

multiplying the temperature-scaled Cross-Entropy loss by a focal term that depends on $p$ does not achieve this, because the gradients would propagate differently through $p$ and $p^\beta$. The desired effect can be attained by using a detached version of $g(p, \gamma)$, which ensures that the gradient is scaled correctly. Therefore, in this work, we present Tempered Focal (TOFU) training objective, which naturally combines the advantages of Focal Loss and GEM.

**Definition 3.4** (TOFU loss). Let us denote $p^\beta$ as a temperature-scaled distribution (2.1), then

$$\mathcal{L}_{TOFU}(\theta) = -\mathbb{E}_q\left[g(\hat{p}_\theta, \gamma)\beta\log p_\theta^\beta\right], \qquad (8)$$

where $g(\hat{p}_\theta, \gamma)$ is detached from gradient computation.

Due to the inclusion of the focal term, TOFU does not directly optimize a well-defined statistic (unlike Cross-Entropy, which minimizes the Kullback–Leibler divergence).

**Corollary 3.5.** *If the target distribution $q$ is one-hot, TOFU loss gradient is proportional to the one of the temperature-scaled CE:*

$$\nabla_\theta \mathcal{L}_{TOFU}(\theta) = g(\hat{p}_\theta, \gamma)\nabla_\theta \mathcal{L}_{CE}^\beta(\theta) \qquad (9)$$

Nevertheless, it can be interpreted as a form of adaptive gradient scaling of the temperature-scaled Cross-Entropy

loss, emphasizing under-predicted tokens while preventing the learned distribution from becoming overly concentrated (Corollary 3.5). As TOFU interpretability is restricted to one-hot target distributions, its applicability to SFT distillation remains tenuous, a limitation that similarly extends to the use of Focal Loss.

## 4. Experimental Evaluation

### 4.1. Tested LLMs

To properly evaluate our method and its alternatives, we aim for an experimental setup that is both realistic and diverse in terms of the models tested. However, ensuring a fair assessment requires careful consideration: many studies on Supervised Fine-Tuning of LLMs rely on models that have already undergone alignment, typically via RLHF (Ouyang et al., 2022) or DPO (Rafailov et al., 2023), which can potentially obscure the isolated effects of SFT. In practice, widely used families such as Llama-3 (Grattafiori et al., 2024), Phi-4 (Abdin et al., 2024), and Qwen-3 (Yang et al., 2025) exhibit security guardrails that suggest an underlying post-training process, though their exact development pipelines and safety mechanisms remain largely opaque. To isolate the impact of SFT itself, we therefore focus on the models that have completed only the pretraining stage, selecting OLMo-2-13B (Groeneveld et al., 2024), Mistral-12B (Mistral AI & NVIDIA, 2024), and Pythia-12B (Biderman et al., 2023). At the same time, to ensure our evaluation remains relevant to real-world usage and captures the diversity of alignment strategies, we also include equal amount of post-trained models: Llama-3.1-8B, Qwen-3-8B, and Phi-4-14B. This setup allows us to evaluate SFT both independently and in combination with alignment.

### 4.2. SFT Setup

To fine-tune the selected models, we used the Alpaca instruction dataset (Taori et al., 2023), a widely adopted collection of diverse instructions paired with demonstrations. As an alternative, we also experimented with the UltraFeedback (Cui et al., 2023) alignment dataset. We performed fine-tuning using the QLoRA framework (Dettmers et al., 2023), which applies Low-Rank Adaptation (Hu et al., 2021) to models quantized with 4-bit NormalFloat (NF4). This approach significantly reduces memory footprint and accelerates training without a substantial loss in performance. More technical details can be found in Appendix B.

### 4.3. Benchmarks

Our evaluations target both diversity and quality. Since the tested models are moderately sized and have undergone only supervised instruction tuning, we favor simple tasks over complex reasoning. Furthermore, we prioritize benchmarks

containing open-ended prompts that favor creative synthesis over deterministic accuracy. To satisfy both criteria, we select three benchmarks that emphasize different aspects of model behavior: **Short Stories (SS)**, **Small Prompts (SP)**, and **NoveltyBench (NB)** (Zhang et al., 2025).

**Short Stories** contains brief narrative beginnings that models are tasked to extend. These entries typically consist of a few introductory sentences to trigger text completion. This format aligns with the original pretraining objective and offers a natural way to observe model diversity. **Small Prompts** focuses on instruction-oriented tasks through a collection of straightforward questions. These entries test general knowledge and basic reasoning. They allow us to measure how diversity behaves when the output must satisfy a specific user request. We composed both datasets from open-source materials, with the curation process detailed in Appendix B.

To further challenge the models, we increase the evaluative complexity by employing **NoveltyBench**, a benchmark designed to measure the capacity for generating multiple distinct, high-quality outputs. It utilizes curated prompts and filtered real-world user queries to elicit maximum response diversity.

To ensure that gains in diversity do not compromise core knowledge, we evaluate models on standard factual benchmarks, MMLU (Hendrycks et al., 2020) and ARC (Clark et al., 2018). These datasets assess a broad spectrum of professional and scientific knowledge acquired during pretraining. Together, they offer a rigorous measure of factual accuracy and knowledge retention, providing a necessary counterpoint to our diversity-focused metrics.

### 4.4. Metrics

To evaluate **diversity**, the main focus of our study, we use the widely adopted Self-BLEU metric (Zhu et al., 2018). For each prompt, it compares the words used in each response to those in the others, measuring how much the outputs overlap in wording and phrasing. This makes Self-BLEU a simple and reproducible way to quantify variation. To verify that increased diversity does not degrade outputs **quality**, we use an instruction-tuned LLM Judge (Appendix B.5) to automatically score responses based on coherence, consistency, and fluency. These evaluation approaches are particularly suitable for datasets such as Short Stories and Small Prompts, where outputs consist of lengthy natural-language sequences, and estimating quality is relatively straightforward.

The NoveltyBench dataset provides a framework for evaluating generated outputs along two complementary metrics: **Distinct** and **Utility**. Distinct measures the diversity of outputs, capturing whether they are semantically differ-

*Table 1.* Performance of models across Alpaca SFT objectives on Short Stories and Small Prompts. Diversity (D) is measured via Self-BLEU (0–100), where lower scores are better. Quality (Q) is measured via LLM Judge score (0–5), where higher scores are better. Red indicates quality degradation under $\lambda$-PR loss, blue indicates optimal diversity and quality retention achieved by TOFU.

| Bench | Method | Mistral-12B | | OLMo-2-13B | | Pythia-12B | | Llama-3.1-8B | | Qwen-3-8B | | Phi-4-14B | |
|---|---|---|---|---|---|---|---|---|---|---|---|---|---|
| | | D↓ | Q↑ | D↓ | Q↑ | D↓ | Q↑ | D↓ | Q↑ | D↓ | Q↑ | D↓ | Q↑ |
| SS | **Base** | $11.4_{\pm4.7}$ | $3.9_{\pm0.5}$ | $12.3_{\pm4.6}$ | $3.7_{\pm0.7}$ | $9.3_{\pm3.1}$ | $2.9_{\pm0.6}$ | $11.5_{\pm6.1}$ | $3.5_{\pm0.7}$ | $24.9_{\pm8.9}$ | $2.9_{\pm1.0}$ | $13.8_{\pm6.8}$ | $3.4_{\pm0.9}$ |
| | **CE** | $22.5_{\pm8.3}$ | $4.8_{\pm0.2}$ | $24.5_{\pm9.1}$ | $4.8_{\pm0.2}$ | $23.8_{\pm8.4}$ | $3.7_{\pm0.6}$ | $22.6_{\pm8.5}$ | $4.7_{\pm0.2}$ | $22.3_{\pm7.7}$ | $4.6_{\pm0.3}$ | $24.8_{\pm9.0}$ | $4.8_{\pm0.2}$ |
| | **FL** | $16.8_{\pm5.4}$ | $4.7_{\pm0.2}$ | $15.9_{\pm5.4}$ | $4.7_{\pm0.2}$ | $14.6_{\pm4.3}$ | $3.8_{\pm0.6}$ | $14.3_{\pm4.7}$ | $4.5_{\pm0.3}$ | $16.4_{\pm6.2}$ | $4.6_{\pm0.3}$ | $16.2_{\pm5.4}$ | $4.7_{\pm0.3}$ |
| | **GEM** | $14.3_{\pm6.0}$ | $4.6_{\pm0.3}$ | $13.2_{\pm5.0}$ | $4.6_{\pm0.3}$ | $13.0_{\pm6.2}$ | $3.4_{\pm0.6}$ | $11.5_{\pm4.0}$ | $4.5_{\pm0.3}$ | $12.7_{\pm3.8}$ | $4.5_{\pm0.3}$ | $12.4_{\pm4.1}$ | $4.6_{\pm0.3}$ |
| | **TOFU** | $12.7_{\pm4.7}$ | $4.6_{\pm0.3}$ | $11.9_{\pm4.1}$ | $4.6_{\pm0.3}$ | $10.8_{\pm3.7}$ | $3.6_{\pm0.5}$ | $11.2_{\pm4.2}$ | $4.5_{\pm0.3}$ | $12.7_{\pm3.8}$ | $4.5_{\pm0.3}$ | $12.4_{\pm4.1}$ | $4.6_{\pm0.3}$ |
| | **$\lambda$-PR** | $3.7_{\pm0.7}$ | $2.5_{\pm0.5}$ | $3.9_{\pm0.6}$ | $3.0_{\pm0.4}$ | $4.1_{\pm0.7}$ | $2.1_{\pm0.5}$ | $3.8_{\pm0.7}$ | $2.7_{\pm0.5}$ | $5.0_{\pm1.2}$ | $3.4_{\pm0.4}$ | $4.0_{\pm0.6}$ | $3.1_{\pm0.4}$ |
| SP | **Base** | $12.7_{\pm7.0}$ | $3.8_{\pm0.9}$ | $13.9_{\pm7.2}$ | $3.8_{\pm1.0}$ | $8.4_{\pm3.0}$ | $2.6_{\pm1.0}$ | $12.2_{\pm6.0}$ | $3.5_{\pm0.9}$ | $31.5_{\pm12.5}$ | $3.8_{\pm0.9}$ | $17.6_{\pm9.3}$ | $3.9_{\pm0.9}$ |
| | **CE** | $44.5_{\pm14.5}$ | $4.2_{\pm0.7}$ | $45.0_{\pm14.6}$ | $4.3_{\pm0.7}$ | $35.9_{\pm14.4}$ | $3.8_{\pm0.8}$ | $44.5_{\pm14.4}$ | $4.2_{\pm0.7}$ | $44.9_{\pm13.4}$ | $4.1_{\pm0.7}$ | $46.6_{\pm15.0}$ | $4.2_{\pm0.7}$ |
| | **FL** | $29.3_{\pm11.1}$ | $4.1_{\pm0.7}$ | $29.0_{\pm9.8}$ | $4.2_{\pm0.7}$ | $22.2_{\pm9.8}$ | $3.6_{\pm0.8}$ | $28.3_{\pm10.5}$ | $4.1_{\pm0.7}$ | $29.2_{\pm10.6}$ | $4.0_{\pm0.7}$ | $28.9_{\pm10.4}$ | $4.2_{\pm0.7}$ |
| | **GEM** | $27.7_{\pm12.6}$ | $4.1_{\pm0.7}$ | $27.0_{\pm11.6}$ | $4.1_{\pm0.7}$ | $17.6_{\pm7.6}$ | $3.5_{\pm0.7}$ | $25.9_{\pm12.3}$ | $4.1_{\pm0.7}$ | $29.5_{\pm11.5}$ | $4.0_{\pm0.8}$ | $27.5_{\pm12.6}$ | $4.1_{\pm0.7}$ |
| | **TOFU** | $21.2_{\pm8.8}$ | $4.1_{\pm0.7}$ | $21.3_{\pm8.6}$ | $4.1_{\pm0.7}$ | $16.1_{\pm7.6}$ | $3.5_{\pm0.8}$ | $20.9_{\pm8.4}$ | $4.0_{\pm0.7}$ | $24.1_{\pm8.9}$ | $4.0_{\pm0.7}$ | $21.8_{\pm8.7}$ | $4.1_{\pm0.6}$ |
| | **$\lambda$-PR** | $2.8_{\pm0.8}$ | $2.0_{\pm0.7}$ | $3.2_{\pm0.9}$ | $2.3_{\pm0.7}$ | $3.0_{\pm1.1}$ | $2.0_{\pm0.6}$ | $2.8_{\pm0.8}$ | $2.1_{\pm0.7}$ | $4.6_{\pm2.1}$ | $2.6_{\pm0.7}$ | $3.3_{\pm1.0}$ | $2.4_{\pm0.6}$ |

ent, while Utility quantifies their quality. Both metrics rely on purposely fine-tuned LLM Judges: one to determine if outputs are meaningfully distinguishable and another to score their usefulness. By focusing on semantics rather than surface-level linguistic variation, NB captures meaningful distinctions between outputs that standard diversity metrics often miss.

## 5. Results

We compare the proposed TOFU loss against the standard Cross-Entropy and the diversity-oriented objectives GEM and $\lambda$-PR. To isolate the impact of prioritizing infrequent patterns, we also evaluate Focal Loss. Hyperparameters settings for GEM and $\lambda$-PR match their original publications, whereas the configurations for Focal Loss and TOFU were derived from ablation studies (see Appendix C).

**Short Stories and Small Prompts**  Table 1 reports the results for Short Stories and Small Prompts using Alpaca SFT. On both benchmarks, Cross-Entropy increases quality but causes a measurable reduction in diversity compared to the base model, confirming the problem addressed in this work. The sole exception is the Qwen-3-8B model when evaluated on Short Stories. Its post-training procedures appear to have a significant residual effect: even in the absence of an instruction template, the model consistently attempts to engage in chain-of-thought reasoning. This case highlights the importance of utilizing unaligned models for SFT-related evaluations to avoid the confounding effects of

prior alignment.

Beyond the initial baseline, our evaluation reveals a clear hierarchy of performance over the benchmarks. Focal Loss yields noticeably higher diversity than Cross-Entropy, though this gain sometimes comes at a slight cost to quality. This validates our hypothesis that diversity is negatively affected by the ignorance of underrepresented training samples during the SFT process. The absolute highest diversity is reached by $\lambda$-PR, but at the expense of quality, pushing the method beyond the bounds of usability. TOFU maintains quality on par with GEM while achieving superior diversity and therefore emerges as the optimal choice. The results using UltraFeedback SFT follow a similar pattern on both benchmarks, although with a general reduction in quality compared to Alpaca (Table D.1, Appendix D).

**NoveltyBench**  For the NoveltyBench evaluation (see Figure 3), the performance hierarchy remains largely consistent with earlier findings. TOFU performs slightly better than the alternative objectives in terms of diversity, with the sole exception of $\lambda$-PR. However, as observed in previous benchmarks, $\lambda$-PR again fails to produce acceptable results, as it generates low-quality outputs that effectively invalidate its high diversity score. The remaining methods perform on par with one another regarding coherence. This parity is particularly evident when accounting for the high variance in the Utility metric across different prompts. Consequently, TOFU represents the most effective solution for this benchmark.

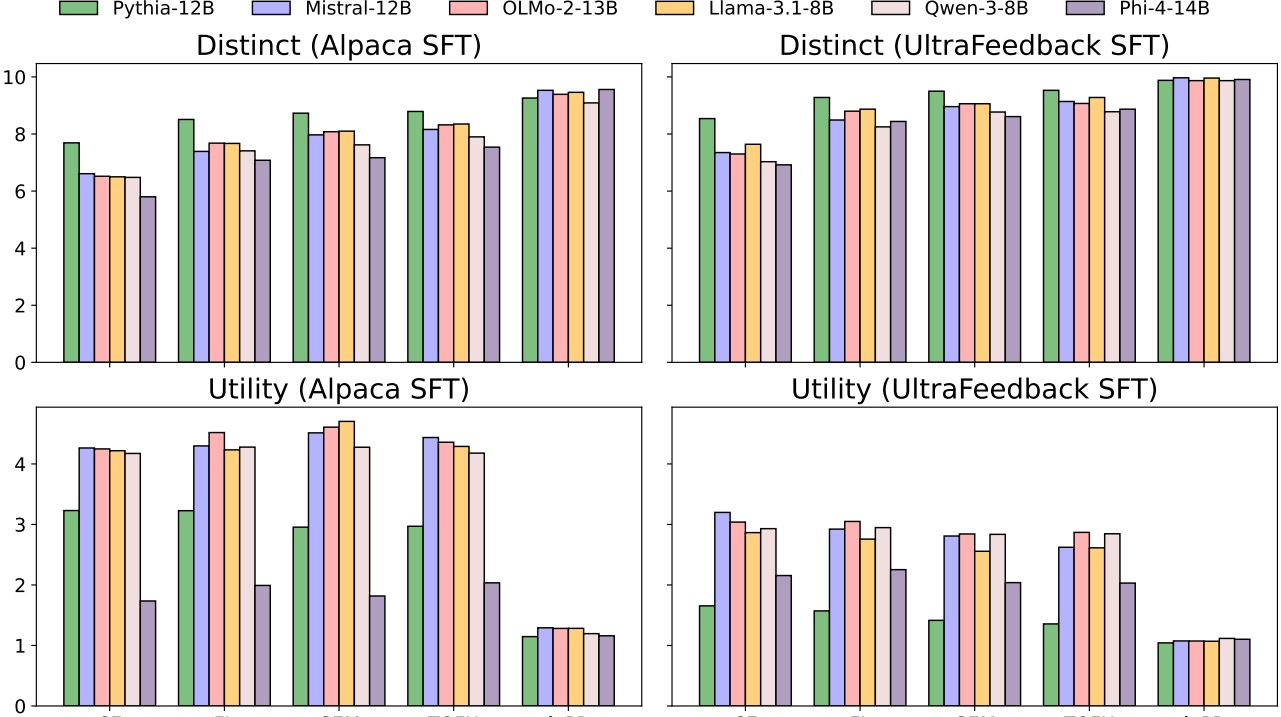

*Figure 3.* The results on NoveltyBench dataset across different models and methods. Distinct (1–10) measures responses diversity, while Utility (1–10) represents quality. For both metrics, higher values indicate superior performance. Error bars (standard deviation) are omitted to maintain visual clarity across multiple model comparisons.

**ARC and MMLU** We further evaluated the models on reasoning-centric benchmarks, such as ARC and MMLU (Figure 4), which probe factual knowledge and reasoning abilities acquired during pretraining. Ultimately, we observe no evidence of catastrophic forgetting across most loss functions, with performance remaining largely comparable to the baseline models. The sole exception is $\lambda$-PR, which consistently exhibits significant performance degradation across all tasks.

Notably, aligned models like Qwen-3-8B and Phi-4-14B exhibit the least degradation under $\lambda$-PR fine-tuning. This suggests that prior post-training may obscure the influence of additional SFT, reinforcing the necessity of using unaligned models to isolate the specific impact of a loss function.

## 6. Conclusion

In this work, we have presented an extensive evaluation of SFT objectives across a range of model families, datasets, and benchmarks. Our investigation into Focal Loss demonstrated its effectiveness as an SFT objective for maintaining the balanced treatment of rare samples. These insights led

to the development of TOFU, a novel training objective designed to mitigate both ignorance and forgetting. Our experimental results confirm that this method achieves superior generative diversity while maintaining competitive output quality compared to standard approaches.

Through the extensive evaluation we provide robust confirmation that standard Cross-Entropy SFT consistently reduces diversity. To our best knowledge, this is the first study to systematically validate this trend across a broad range of model scales and diverse SFT datasets, as previous work of O'Mahony et al. (2024) primarily explored this hypothesis within the Pythia family at smaller sizes. By extending the evaluation to these more substantial architectures, we provide empirical evidence that diversity loss is an inherent characteristic of the standard SFT pipeline rather than an artifact of specific smaller-scale architectures.

We initially attributed the decline to two primary factors: the forgetting of pretrained information and the ignorance of low-frequency patterns in the fine-tuning data. This hypothesis is supported by the performance of GEM, designed to mitigate forgetting, and Focal Loss, designed to prioritize underrepresented samples, both of which yield measurable

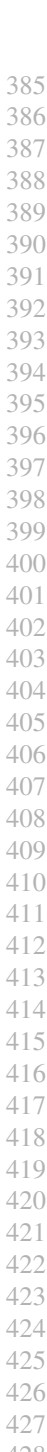
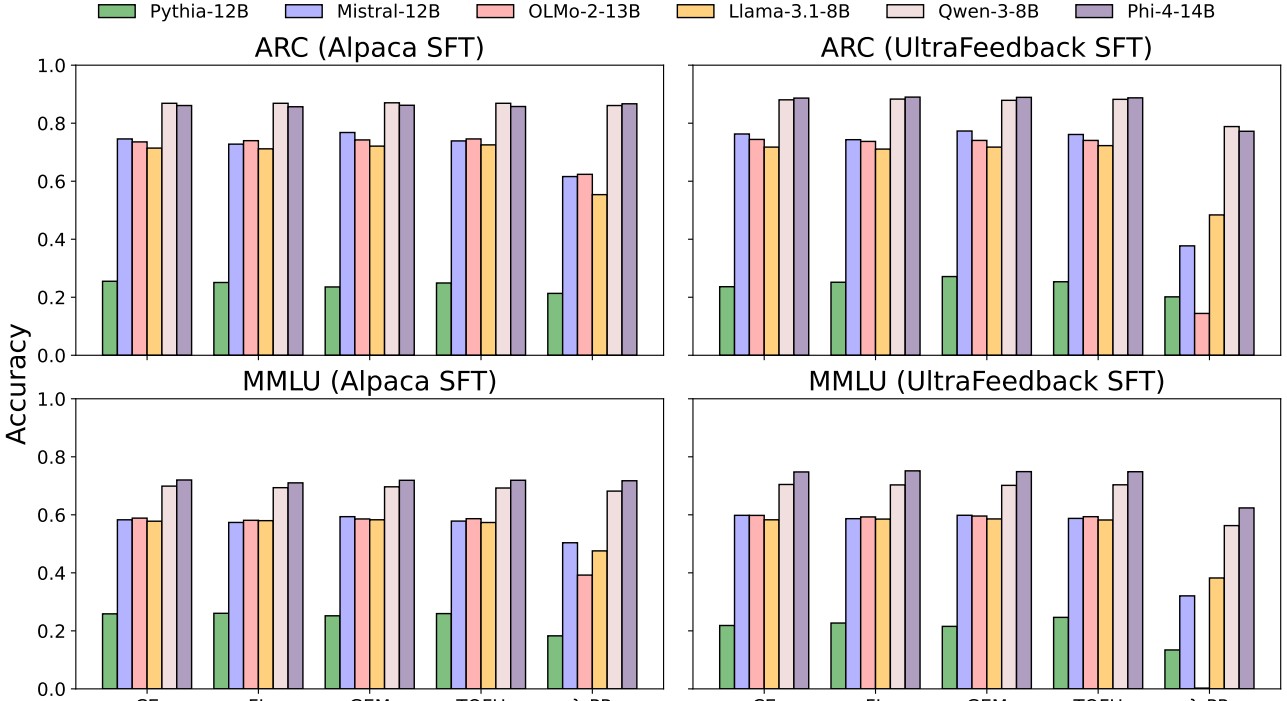

*Figure 4.* Evaluation results for the ARC and MMLU benchmarks across the tested models and fine-tuning methods. Accuracy reflects the percentage of correct predictions made by the model. The values represent the mean accuracy scores across all tasks in each benchmark.

improvements in diversity compared to CE.

The proposed TOFU loss function, designed to mitigate both forgetting and ignorance, achieves the highest diversity while maintaining highly competitive quality across our benchmarks. These results suggest that a principled theoretical approach to loss function design is broadly transferable to various language modeling applications. We note that in certain evaluations, Focal Loss maintains a slight edge in quality over TOFU, potentially making it a more conservative choice for downstream tasks where quality preservation is the absolute priority.

Furthermore, while TOFU demonstrated stable performance across various model sizes and SFT datasets, there remains potential for further validation. Given greater computational resources, future experiments could evaluate these objectives on larger-scale architectures and higher-quality closed-source instruction tuning datasets. Finally, while this study focuses on the supervised phase, the influence of these diversity-preserving objectives on subsequent Reinforcement Learning stages remains an open question for future investigation. We hope our research encourages further exploration into specialized training objectives inspired by our theoretical approach. Additionally, we look forward

to seeing the community apply more extensive resources to test and build upon these findings.

**Reproducibility Statement** The code and datasets are available at https://github.com/anonymous2026icml/TOFU. Upon publication, the source code will be made available on GitHub, while the model weights and datasets will be hosted on Hugging Face.

## Impact Statement

This work aims to advance the field of Machine Learning by addressing generative diversity and knowledge retention in fine-tuned Large Language Models. By mitigating the forgetting of pretrained information and the underrepresentation of rare patterns, our research supports the development of models that produce more varied and information-rich outputs. There are many potential societal consequences of improving generative variety, none of which we feel must be specifically highlighted here beyond the ethical considerations standard to the advancement of language modeling.

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

## Appendix

## A. Omitted proofs

**Proposition A.1** (Notation and Logit Gradients)**.** *We introduce the notation used throughout the following proofs and derivations. All expressions below are written for a single token, with indices ranging over the vocabulary (i.e., number of classes). Consider a temperature-scaled distribution $p^\beta$ from Definition (2.1), a Kronecker's delta matrix $\delta_{ij}$, and $z$ as model's output logits. Let us denote the following log probabilities*

$$l_i = \log p_i = z_i - \log \sum_k e^{z_k} \tag{A.1}$$

$$l_i^\beta = \log p_i^\beta = \frac{z_i}{\beta} - \log \sum_k e^{\frac{z_k}{\beta}} \tag{A.2}$$

*Then their derivatives with respect to the logits are given by*

$$\frac{\partial l_i}{\partial z_j} = \delta_{ij} - p_j = \delta_{ij} - e^{l_j} \tag{A.3}$$

$$\frac{\partial l_i^\beta}{\partial z_j} = \frac{1}{\beta}(\delta_{ij} - p_j^\beta) \tag{A.4}$$

*Proof.* The computation is straightforward, partial derivatives of log probabilities are computed as

$$\frac{\partial l_i}{\partial z_j} = \delta_{ij} - \frac{e^{z_j}}{\sum_k e^{z_k}} = \delta_{ij} - e^{l_j} = \delta_{ij} - p_j, \tag{A.5}$$

$$\frac{\partial l_i^\beta}{\partial z_j} = \frac{1}{\beta}\delta_{ij} - \frac{1}{\beta}\frac{e^{\frac{z_j}{\beta}}}{\sum_k e^{\frac{z_k}{\beta}}} = \frac{1}{\beta}(\delta_{ij} - p_j^\beta). \tag{A.6}$$

$\square$

While the final loss gradient with respect to the model parameters is independent of whether intermediate derivatives are computed via $l$ or $p$, practical implementations that omit the explicit calculation of $p$ make derivatives with respect to $l$ more informative. We illustrate this on the Shannon entropy example:

$$\mathcal{H} \doteq -p_i \log p_i = -p_i l_i. \tag{A.7}$$

Its gradients with respect to $p$ are

$$\frac{\partial \mathcal{H}}{\partial p_i} = \frac{\partial}{\partial p_i}(-p_i \log p_i) = -\log p_i - 1. \tag{A.8}$$

Based on the eq. (A.8) in low probability setting when $p \to 0$ the Shannon entropy gradient diverges $\frac{\partial \mathcal{H}}{\partial p_i} \to -\infty$. We now examine the gradients with respect to log probabilities, as these are the values calculated in practical implementations:

$$\frac{\partial \mathcal{H}}{\partial l_i} = \frac{\partial}{\partial l_i}(-p_i l_i) = -l_i \frac{\partial p_i}{\partial l_i} - p_i = -l_i p_i - p_i = -p_i \log p_i - p_i. \tag{A.9}$$

From the eq. (A.9) it is clear that there is no explosion as when $p \to 0$, the gradients are $\frac{\partial \mathcal{H}}{\partial l_i} \to 0$ due to the following limit:

$$\lim_{p \to 0}(lp - p) = \lim_{p \to 0}(p \log p - p) = 0. \tag{A.10}$$

The gradients remain robust throughout the rest of backpropagation chain:

$$\frac{\partial \mathcal{H}}{\partial z_j} = \sum_i \frac{\partial \mathcal{H}}{\partial l_i}\frac{\partial l_i}{\partial z_j} = \sum_i -(l_i p_i + p_i)(\delta_{ij} - p_j) = -l_j p_j + \sum_i l_i p_i p_j = -p_j\left(l_j - \sum_i l_i p_i\right) \tag{A.11}$$

This example demonstrates how using $l$ rather than $p$ helps avoid false assumptions regarding the method's behavior.

Proposition A.2 derives the formulation of the GEM loss used in Definition 2.1, translating the original objective into our notation and providing a simplified expression that is more comprehensible on sight.

**Proposition A.2** (Reformulation of the original GEM from (Li et al., 2024) with our notation). *Consider a model distribution $p_\theta$, data distribution $q$ and a temperature-scaled distribution $p_\theta^\beta = \text{softmax}(\beta^{-1} \log p_\theta)$. Consider $y^{real}$ is the supervised label in the dataset and $y^{gene}$ is the model-generated output. The original GEM formulation is provided in the following equation:*

$$\mathcal{L}_{GEM}(\theta) = \sum_{y^{real} \sim q} \sum_{y^{gene} \sim p_\theta} p_\theta^\beta(y^{gene}) \left[ \log p_\theta(y^{gene}) - \log p_\theta(y^{real}) \right], \tag{A.12}$$

*where $\beta \in (0, 1)$ is a temperature parameter and $p^\beta(y^{gene})$ is detached from gradient computational graph. The loss (A.12) in our notation can be reformulated as*

$$\mathcal{L}_{GEM}(\theta) = -\mathbb{E}_q[\log p_\theta] + \mathbb{E}_{p_\theta^\beta}[\log p_\theta]. \tag{A.13}$$

*Proof.* Let us rewrite the eq. (A.12) with respect to the detached term:

$$\mathcal{L}_{GEM}(\theta) = \mathbb{E}_{y^{real} \sim q} \mathbb{E}_{y^{gene} \sim p_\theta^\beta} \left[ \log p_\theta(y^{gene}) - \log p_\theta(y^{real}) \right] = \tag{A.14}$$

$$= \mathbb{E}_{y^{real} \sim q} \mathbb{E}_{y^{gene} \sim p_\theta^\beta} \left[ \log p_\theta(y^{gene}) \right] - \mathbb{E}_{y^{real} \sim q} \mathbb{E}_{y^{gene} \sim p_\theta^\beta} \left[ \log p_\theta(y^{real}) \right] = \tag{A.15}$$

$$= \mathbb{E}_{y^{gene} \sim p_\theta^\beta} \left[ \log p_\theta(y^{gene}) \right] - \mathbb{E}_{y^{real} \sim q} \left[ \log p_\theta(y^{real}) \right]. \tag{A.16}$$

Now, by disregarding notational $y^{gene}$ and $y^{real}$, it is clear that GEM exactly matches the following:

$$\mathcal{L}_{GEM}(\theta) = -\mathbb{E}_q[\log p_\theta] + \mathbb{E}_{p_\theta^\beta}[\log p_\theta]. \tag{A.17}$$

$\square$

**Theorem A.3** (Theorem 3.1). *Training with $\mathcal{L}_{GEM}$ is equivalent to training with temperature-scaled Cross-Entropy loss, as*

$$\nabla_\theta \mathcal{L}_{GEM}(\theta) = \nabla_\theta \mathcal{L}_{CE}^\beta(\theta), \tag{A.18}$$

*where $\mathcal{L}_{CE}^\beta(\theta) \doteq -\beta \mathbb{E}_q \log p_\theta^\beta$.*

*Proof.* The Cross-Entropy loss is defined as $\mathcal{L}_{CE} = -q_i \log p_i = -q_i l_i$. Using the chain rule and Proposition A.1, Cross-Entropy gradients with respect to log probabilities and logits are

$$\frac{\partial \mathcal{L}_{CE}}{\partial l_i} = -q_i, \tag{A.19}$$

$$\frac{\partial \mathcal{L}_{CE}}{\partial z_j} = \sum_i \frac{\partial \mathcal{L}_{CE}}{\partial l_i} \frac{\partial l_i}{\partial z_j} = -\sum_i q_i(\delta_{ij} - p_j) = p_j - q_j. \tag{A.20}$$

Similarly, gradients for $\mathcal{L}_{GEM} = -q_i \log p_i + p_i^\beta \log p_i$ are

$$\frac{\partial \mathcal{L}_{GEM}}{\partial z_j} = \sum_i \frac{\partial \mathcal{L}_{GEM}}{\partial l_i} \frac{\partial l_i}{\partial z_j} = \sum_i (p_i^\beta - q_i)(\delta_{ij} - p_j) = p_j^\beta - q_j. \tag{A.21}$$

Let us calculate the gradients for scaled Cross-Entropy $\mathcal{L}_{CE}^\beta = -\beta q_i \log p_i^\beta = -\beta q_i l_i^\beta$.

$$\frac{\partial \mathcal{L}_{CE}^\beta}{\partial z_j} = \sum_i \frac{\partial \mathcal{L}_{CE}^\beta}{\partial l_i^\beta} \frac{\partial l_i^\beta}{\partial z_j} = -\sum_i \beta q_i \cdot \frac{1}{\beta}(\delta_{ij} - p_j^\beta) = p_j^\beta - q_j. \tag{A.22}$$

Now, as $\nabla_\theta z$ does not depend on the loss function, we have

$$\nabla_\theta \mathcal{L}_{GEM}(\theta) = \nabla_\theta \mathcal{L}_{CE}^\beta(\theta). \tag{A.23}$$

$\square$

**Proposition A.4** (Proposition 3.2)**.** *Assume that the target distribution $q$ is one-hot. Then, for Focal Loss $\mathcal{L}_{FL}$ and Cross-Entropy $\mathcal{L}_{CE}$, the gradients satisfy*

$$\nabla_\theta \mathcal{L}_{FL}(\theta) = g(\hat{p}_\theta, \gamma)\nabla_\theta \mathcal{L}_{CE}(\theta), \tag{A.24}$$

*where $g(p, \gamma) = (1-p)^\gamma - \gamma p(1-p)^{\gamma-1}\log p$. Here and further $\hat{p}$ denotes the predicted probability assigned to the ground-truth token.*

*Proof.* First let's consider $q$ to be arbitrary. Then by the chain rule and Proposition A.1 the gradients of Focal Loss are

$$\frac{\partial \mathcal{L}_{FL}}{\partial l_i} = -(1-p_i)^\gamma q_i + \gamma(1-p_i)^{\gamma-1}q_i l_i p_i, \tag{A.25}$$

$$\frac{\partial \mathcal{L}_{FL}}{\partial z_j} = \sum_i \frac{\partial \mathcal{L}_{FL}}{\partial l_i}\frac{\partial l_i}{\partial z_j} = -(1-p_j)^\gamma q_j + \gamma(1-p_j)^{\gamma-1}q_j l_j p_j + \sum_i (1-p_i)^\gamma q_i p_j - \gamma(1-p_i)^{\gamma-1}q_i l_i p_i p_j = \tag{A.26}$$

$$= -q_j((1-p_j)^\gamma - \gamma(1-p_j)^{\gamma-1}l_j p_j) + p_j \sum_i ((1-p_i)^\gamma q_i - \gamma(1-p_i)^{\gamma-1}q_i l_i p_i). \tag{A.27}$$

In this case, the resulting gradients are not proportional to the gradients of the Cross-Entropy loss. However, if $q$ is one-hot, meaning $q_k = 1$ and $q_{i \neq k} = 0$ for some $k$, then

$$\frac{\partial \mathcal{L}_{FL}}{\partial z_j} = \underbrace{((1-p_k)^\gamma - \gamma(1-p_k)^{\gamma-1}l_k p_k)}_{g(p_k, \gamma)}(p_j - q_j), \tag{A.28}$$

and therefore, as $\nabla_\theta z$ does not depend on the loss function and denoting $(1-p)^\gamma - \gamma p(1-p)^{\gamma-1}\log p$ as $g(p, \gamma)$, we have

$$\nabla_\theta \mathcal{L}_{FL}(\theta) = g(\hat{p}_\theta, \gamma)\nabla_\theta \mathcal{L}_{CE}(\theta). \tag{A.29}$$

$\square$

**Corollary A.5** (Corollary 3.3)**.** *If the target distribution $q$ is one-hot*

$$-\beta\nabla_\theta \mathbb{E}_q\left[(1-p_\theta^\beta)^\gamma \log p_\theta^\beta\right] = g(\hat{p}_\theta^\beta, \gamma)\nabla_\theta \mathcal{L}_{CE}^\beta(\theta) \tag{A.30}$$

*Proof.* By the chain rule, Proposition A.1, and analogously to the proof of Proposition 3.2, if the target distribution $q$ is one-hot

$$\frac{\partial}{\partial z_j}(1-p_i^\beta)^\gamma q_i \log p_i^\beta = \underbrace{((1-p_k^\beta)^\gamma - \gamma(1-p_k^\beta)^{\gamma-1}l_k^\beta p_k^\beta)}_{g(\hat{p}_k^\beta, \gamma)}(q_j - p_j^\beta). \tag{A.31}$$

Therefore, as $\nabla_\theta z$ does not depend on the loss function

$$-\beta\nabla_\theta \mathbb{E}_q\left[(1-p_\theta^\beta)^\gamma \log p_\theta^\beta\right] = g(\hat{p}_\theta^\beta, \gamma)\nabla_\theta \mathcal{L}_{CE}^\beta(\theta). \tag{A.32}$$

$\square$

**Corollary A.6** (Corollary 3.5)**.** *If the target distribution $q$ is one-hot, TOFU gradients are proportional to the ones of the temperature-scaled CE:*

$$\nabla_\theta \mathcal{L}_{TOFU}(\theta) = g(\hat{p}_\theta, \gamma)\nabla_\theta \mathcal{L}_{CE}^\beta(\theta) \tag{A.33}$$

*Proof.* As $g(\hat{p}_\theta, \gamma)$ is detached from gradients computation

$$\nabla_\theta \mathcal{L}_{TOFU}(\theta) = \nabla_\theta g(\hat{p}_\theta, \gamma)\mathcal{L}_{CE}^\beta(\theta) = g(\hat{p}_\theta, \gamma)\nabla_\theta \mathcal{L}_{CE}^\beta(\theta) \tag{A.34}$$

$\square$

## B. Experimental details

In this section we provide comprehensive descriptions for the models, benchmarks and datasets used in our study.

### B.1. Models

**OLMo-2-13B** OLMo-2-1124-13B (Groeneveld et al., 2024) is a fully open-source language model from the Allen Institute for AI, designed for research and educational use. It is trained on the Dolma (Soldaini et al., 2024) dataset for improved performance on tasks such as text generation and instruction following. The model is part of the OLMo series, which emphasizes transparency by releasing weights, data, and training details.

**Mistral-12B** Mistral NeMo (Mistral AI & NVIDIA, 2024) is a high-performance pretrained language model trained jointly by Mistral AI and NVIDIA. While it features a significantly extended context window, many of its specific training details remain undisclosed. It is designed for diverse tasks including text generation and instruction following.

**Pythia-12B** Pythia 12B (Biderman et al., 2023) belongs to a suite of 16 models trained on the Pile (Gao et al., 2020), a diverse dataset comprising academic, web, and dialogue content. With 12 billion parameters, it is specifically designed as a scientific tool for studying model functionality and interpretability rather than for deployment or human-facing interactions. This controlled training setup, where models of varying sizes share the same data and hyperparameters, makes it ideal for the comparative analysis of training objectives.

**Llama-3.1-8B** Llama-3.1 (Touvron et al., 2023) is a large-scale open-weight language model released by Meta AI as an extension of the Llama-3 series. It is trained on a substantially expanded and higher-quality corpus, with improvements in multilingual coverage, reasoning ability, and instruction-following performance. The model serves as a strong foundation for downstream fine-tuning and alignment methods such as SFT and preference optimization, making it widely adopted in both research and applied settings.

**Qwen-3-8B** Qwen3 (Yang et al., 2025) is a family of large language models developed by Alibaba Cloud, designed to support general-purpose language understanding, reasoning, and instruction-following tasks. Built upon the Qwen architecture, Qwen3 incorporates architectural and data-scale improvements over its predecessors, including enhanced multilingual and code capabilities. The model is trained on a diverse mixture of web, code, and domain-specific data. Qwen3 is released with open weights, enabling transparent research and efficient fine-tuning for downstream applications.

**Phi-4-14B** Phi-4 (Abdin et al., 2024) is a compact yet high-performing language model family developed by Microsoft, focusing on efficiency and strong reasoning capabilities under limited parameter budgets. It is trained using a carefully curated dataset that emphasizes high-quality, synthetic, and textbook-style data.

### B.2. SFT Datasets

**Alpaca** Alpaca[1] (Taori et al., 2023) is a widely used instruction-following dataset consisting of approximately 52K instruction–response pairs generated using a self-instruct framework. The dataset covers a broad range of tasks, including question answering, summarization, reasoning, and creative writing. Alpaca is commonly used for supervised fine-tuning to adapt base language models toward instruction-following behavior. To preprocess the Alpaca dataset, we filter and format each example into prompt–completion pairs. We use explicit delimiters for the instruction, input, and response to provide structural context for the sequence. The model is trained to generate the response following an opening delimiter and is explicitly required to produce a matching delimiter to signal completion.

**UltraFeedback-Binarized** UltraFeedback-Binarized (Cui et al., 2023) is a preference-annotated dataset hosted on Hugging Face by openbmb[2], derived from the UltraFeedback corpus and adapted for supervised fine-tuning and instruction tuning. The dataset comprises 64K samples, each containing an AI-generated judgment that identifies a preferred response. These preference labels facilitate supervised training, allowing models to learn instruction-following behavior and high-quality response patterns. Our pipeline filters the UltraFeedback dataset for top-quality examples, retaining only the highest-scoring

---

[1]https://huggingface.co/datasets/tatsu-lab/alpaca
[2]https://huggingface.co/datasets/openbmb/UltraFeedback

completion per prompt, provided that it meets a minimum score of 7, resulting in 57,400 samples. We format these pairs using fixed templates the same as for the Alpaca dataset.

### B.3. SFT details

Given the limitation of our computational resources, we performed 4-bit NormalFloat quantization of selected models and utilized the Quantized Low Rank Adaptation (Dettmers et al., 2023) technique to optimize our workflow. On top of that, we used gradient accumulation to increase the total batch size. The exact training parameters such as learning rate, batch size, LoRA ranks, amount of gradient accumulation steps are explicitly provided in our source code. We fine-tune all models for a single epoch to mitigate memorization of relatively small instruction datasets such as Alpaca and UltraFeedback.

### B.4. Evaluation Datasets

**General inference information**     The model's generation parameters were selected based on the specific requirements of each evaluation task. Following standard empirical practices in the field, we employed a stochastic sampling strategy for the Short Stories, Small Prompts, and NoveltyBench datasets. Specifically, we used nucleus sampling (Holtzman et al., 2020) with a cumulative probability threshold of $p = 0.9$ and a unit temperature $T = 1.0$. For these benchmarks we generated 10 responses per prompt. In contrast, for the ARC and MMLU, we utilized deterministic greedy decoding to ensure objective and reproducible outputs, generating a single completion per prompt.

Due to computational constraints, the maximum response length was restricted across all experiments. We allocated a limit of 64 tokens for generative benchmarks and 8 tokens for multiple-choice ones.

**Short Stories**     Short Stories is a continuation benchmark where a model is given a story beginning and must generate a coherent conclusion. We observed that base models faithfully continue the narrative when provided with a substantial initial context. Since this means that the task does not require an explicit instruction format, it allows for direct comparison between base models and their fine-tuned counterparts. For the SFT models, we adhere to the standard instruction template, incorporating a prompt that mandates story continuation to ensure the intended inference mode.

We constructed this dataset by randomly selecting 100 different stories from the ROCStories[3] corpus (Mostafazadeh et al., 2016) , each containing exactly five sentences. For our evaluation, the first four sentences serve as the beginning of the story, providing sufficient context for a logical continuation.

**Small Prompts**     The Small Prompts dataset is a collection of short questions curated from the original AlpacaFarm (Dubois et al., 2023) Hugging Face repository[4]. For SFT models, we utilize the standard instruction template without supplementary prompting. For base models, we follow the prompting protocol established by O'Mahony et al. (2024).

To ensure the evaluation remains focused on standard natural language, we extracted a *helpful_base* subset of 129 diverse prompts from the initial 805 entries. This selection excludes non-prose outputs, such as code or ASCII symbol drawings. We restrict the evaluation to natural language because the Self-BLEU metric does not function reliably outside of this domain, as its sensitivity to structural repetition in code or symbols can produce misleading diversity scores.

**NoveltyBench**     NoveltyBench (Zhang et al., 2025) is a benchmark designed to evaluate language models' ability to generate multiple distinct and high-quality outputs for the same prompt, addressing limitations of traditional benchmarks that focus on a single best response. Originally, the benchmark provides two distinct datasets: *NB-curated* which contains 100 prompts manually curated by the authors and *NB-WildChat* which consists of 1,000 prompts automatically curated from real user interactions with ChatGPT.

In our work, we selected *NB-curated* subset for the final evaluation. We utilized the original NoveltyBench framework and source code, including the default parameters for their proprietary quality and diversity metrics, Utility-k and Distinct-k.

**MMLU**     Massive Multitask Language Understanding (MMLU) (Hendrycks et al., 2020) is a comprehensive benchmark designed to evaluate the knowledge and reasoning capabilities of language models across multiple subject areas, spanning STEM disciplines, humanities, and social sciences. The dataset includes questions of varying difficulty levels, ranging from

---

[3]https://huggingface.co/datasets/mintujupally/ROCStories
[4]https://huggingface.co/datasets/tatsu-lab/alpaca_farm

elementary concepts to advanced professional knowledge. For the benchmarking, we used its test subset, comprising 14042 questions. We converted the dataset's answer indices into explicit labels (A–D). Models were prompted to output a single choice label in a predefined textual format. We calculate the final accuracy against the ground-truth labels.

**ARC**    ARC-Challenge (ARC) (Clark et al., 2018) is a benchmark dataset of multiple-choice science questions curated to evaluate advanced reasoning and scientific understanding. The questions are sourced from standardized science examinations for grades 3 through 9 and are intentionally selected to be challenging for both humans and AI systems. For the benchmarking, we used its test subset, comprising 1172 questions. While MMLU answers are indexed implicitly and mapped to canonical labels (A–D), ARC provides explicit choice labels that are preserved during preprocessing. We calculate the final accuracy against the ground-truth labels.

### B.5. LLM Judge

To access the quality of the responses, we employ large language model as a judge to score and compare generated responses. Specifically, we use **Llama-3.1-70B-Instruct** (Grattafiori et al., 2024), an instruction-aligned model, as complementary evaluator. To reduce memory footprint, we quantized the Judge with 4-bit NormalFloat. Ultimately, we used LLM Judge to score responses on Short Stories and Small Prompts datasets.

For both datasets we prompted the Judge with comprehensive instructions to provide a score from 0 to 5, where 0 corresponds to an incoherent, off-topic, or nonsensical response, and 5 corresponds to a seamless, natural, and stylistically consistent response. Given the restriction in tokens that we applied in the inference stage, we explicitly state in the Judge instruction not to penalize the response if it ends abruptly due to the aforementioned limit. However, it is required to penalize a logically or stylistically flawed ending. To ensure a robust evaluation, we used greedy decoding and strict response template.

## C. Ablations

We find best parameters for Focal Loss and TOFU objectives by evaluating quality and diversity on NoveltyBench and accuracy on ARC. For these experiments we employ Mistral-12B model. We tested the following ranges of parameters: $\gamma \in [2, 5]$ and $\beta \in [0.6, 0.9]$. According to the results, gathered in Table C.1, the best hyperparameters are $\gamma = 3$ for Focal Loss and $\gamma = 3, \beta = 0.8$ for TOFU. While we selected the optimal values primarily based on ARC accuracy, they also coincide with the top-performing configurations for NoveltyBench Utility.

*Table C.1.* ARC Accuracy, NoveltyBench Distinct (1-10) and Utility (1-10) results for the Mistral-12B model fine-tuned with Focal Loss and TOFU with different $(\gamma, \beta)$ configurations. The optimal choices are highlighted in blue.

| | **Focal Loss** | | | | **TOFU** | | | | | | | | | | | |
|---|---|---|---|---|---|---|---|---|---|---|---|---|---|---|---|---|
| $\beta$ | - | - | - | - | 0.6 | 0.6 | 0.6 | 0.7 | 0.7 | 0.7 | 0.8 | 0.8 | 0.8 | 0.9 | 0.9 | 0.9 |
| $\gamma$ | 2 | 3 | 4 | 5 | 2 | 3 | 5 | 2 | 3 | 5 | 2 | 3 | 5 | 2 | 3 | 5 |
| Accuracy ↑ | 72.9 | 75.0 | 74.3 | 74.2 | 72.0 | 74.4 | 75.4 | 73.1 | 73.8 | 74.6 | 75.3 | 75.6 | 74.5 | 71.6 | 74.7 | 74.6 |
| Distinct ↑ | 3.97 | 4.23 | 4.44 | 4.37 | 4.64 | 4.69 | 4.76 | 4.46 | 4.40 | 4.69 | 4.25 | 4.41 | 4.54 | 4.18 | 4.36 | 4.45 |
| Utility ↑ | 4.61 | 4.90 | 4.81 | 4.29 | 4.30 | 4.13 | 3.97 | 4.53 | 4.37 | 4.17 | 4.72 | 4.71 | 4.22 | 4.76 | 4.65 | 4.16 |

## D. Additional results

To verify the robustness of our results across different SFT datasets, we replicated the experiments conducted on Alpaca using UltraFeedback. While the lower overall quality of UltraFeedback negatively affects performance, the trends observed in the Alpaca experiments remain preserved. The corresponding results for Short Stories and Small Prompts are provided in Table D.1.

While the primary figures for NoveltyBench are presented in the main text, the corresponding raw values are provided here in Table D.2. We note that the Utility metric exhibits a high standard deviation, in some cases exceeding half of the mean value. This significant variance suggests the metric may be unstable or less reliable for certain comparisons. This significant variance may stem from the diverse difficulty levels of the prompts within the benchmark, or potentially from a lack of robustness in the judge used for evaluation.

*Table D.1.* Performance of models across UltraFeedback SFT objectives on Short Stories and Small Prompts. Diversity (D) is measured via Self-BLEU (0–100), where lower scores are better. Quality (Q) is measured via LLM Judge score (0–5), where higher scores are better. Red indicates quality degradation under $\lambda$-PR loss, blue indicates optimal diversity and quality retention achieved by TOFU.

| Bench | Method | Mistral-12B | | OLMo-2-13B | | Pythia-12B | | Llama-3.1-8B | | Qwen-3-8B | | Phi-4-14B | |
|---|---|---|---|---|---|---|---|---|---|---|---|---|---|
| | | D↓ | Q↑ | D↓ | Q↑ | D↓ | Q↑ | D↓ | Q↑ | D↓ | Q↑ | D↓ | Q↑ |
| SS | Base | $11.4_{\pm4.7}$ | $3.9_{\pm0.5}$ | $12.3_{\pm4.6}$ | $3.7_{\pm0.7}$ | $9.3_{\pm3.1}$ | $2.9_{\pm0.6}$ | $11.5_{\pm6.1}$ | $3.5_{\pm0.7}$ | $24.9_{\pm8.9}$ | $2.9_{\pm1.0}$ | $13.8_{\pm6.8}$ | $3.4_{\pm0.9}$ |
| | CE | $26.9_{\pm9.8}$ | $4.4_{\pm0.4}$ | $24.0_{\pm8.7}$ | $4.4_{\pm0.5}$ | $19.7_{\pm7.4}$ | $3.1_{\pm0.7}$ | $23.9_{\pm8.8}$ | $4.2_{\pm0.5}$ | $25.6_{\pm12.0}$ | $4.2_{\pm0.5}$ | $24.4_{\pm7.9}$ | $4.5_{\pm0.3}$ |
| | FL | $15.3_{\pm4.7}$ | $4.4_{\pm0.3}$ | $14.3_{\pm4.6}$ | $4.3_{\pm0.4}$ | $14.9_{\pm4.7}$ | $2.7_{\pm0.6}$ | $14.9_{\pm5.4}$ | $4.1_{\pm0.4}$ | $15.1_{\pm5.7}$ | $4.2_{\pm0.4}$ | $15.5_{\pm5.4}$ | $4.4_{\pm0.3}$ |
| | GEM | $13.1_{\pm5.3}$ | $4.2_{\pm0.4}$ | $12.1_{\pm4.3}$ | $4.1_{\pm0.5}$ | $9.8_{\pm3.6}$ | $2.6_{\pm0.7}$ | $12.0_{\pm4.8}$ | $4.0_{\pm0.4}$ | $13.4_{\pm6.5}$ | $4.1_{\pm0.4}$ | $13.1_{\pm5.8}$ | $4.3_{\pm0.4}$ |
| | TOFU | $12.1_{\pm4.6}$ | $4.2_{\pm0.3}$ | $11.2_{\pm3.5}$ | $4.2_{\pm0.4}$ | $9.5_{\pm3.0}$ | $2.9_{\pm0.6}$ | $11.4_{\pm3.7}$ | $4.0_{\pm0.4}$ | $12.3_{\pm4.7}$ | $4.1_{\pm0.4}$ | $11.8_{\pm4.3}$ | $4.3_{\pm0.4}$ |
| | $\lambda$-PR | $3.3_{\pm0.5}$ | $1.8_{\pm0.5}$ | $3.2_{\pm0.4}$ | $2.1_{\pm0.5}$ | $3.6_{\pm0.7}$ | $1.2_{\pm0.5}$ | $3.3_{\pm0.5}$ | $1.9_{\pm0.5}$ | $4.0_{\pm0.8}$ | $2.6_{\pm0.5}$ | $3.4_{\pm0.5}$ | $2.2_{\pm0.5}$ |
| SP | Base | $12.7_{\pm7.0}$ | $3.8_{\pm0.9}$ | $13.9_{\pm7.2}$ | $3.8_{\pm1.0}$ | $8.4_{\pm3.0}$ | $2.6_{\pm1.0}$ | $12.2_{\pm6.0}$ | $3.5_{\pm0.9}$ | $31.5_{\pm12.5}$ | $3.8_{\pm0.9}$ | $17.6_{\pm9.3}$ | $3.9_{\pm0.9}$ |
| | CE | $45.1_{\pm10.3}$ | $3.8_{\pm0.5}$ | $43.9_{\pm9.6}$ | $3.8_{\pm0.6}$ | $32.0_{\pm9.0}$ | $3.2_{\pm0.6}$ | $42.8_{\pm10.0}$ | $3.8_{\pm0.5}$ | $44.3_{\pm9.8}$ | $3.7_{\pm0.6}$ | $48.0_{\pm9.9}$ | $3.8_{\pm0.5}$ |
| | FL | $27.8_{\pm7.1}$ | $3.6_{\pm0.5}$ | $26.8_{\pm7.0}$ | $3.6_{\pm0.5}$ | $20.6_{\pm5.8}$ | $3.0_{\pm0.5}$ | $26.0_{\pm7.0}$ | $3.6_{\pm0.5}$ | $28.3_{\pm7.5}$ | $3.5_{\pm0.5}$ | $27.9_{\pm7.3}$ | $3.6_{\pm0.5}$ |
| | GEM | $26.3_{\pm8.0}$ | $3.6_{\pm0.5}$ | $25.1_{\pm7.3}$ | $3.5_{\pm0.5}$ | $16.7_{\pm5.7}$ | $2.8_{\pm0.6}$ | $24.2_{\pm6.9}$ | $3.5_{\pm0.5}$ | $28.4_{\pm8.5}$ | $3.5_{\pm0.5}$ | $28.1_{\pm8.0}$ | $3.6_{\pm0.5}$ |
| | TOFU | $20.9_{\pm5.7}$ | $3.5_{\pm0.6}$ | $20.2_{\pm5.7}$ | $3.5_{\pm0.5}$ | $14.6_{\pm4.5}$ | $2.8_{\pm0.6}$ | $19.8_{\pm5.7}$ | $3.5_{\pm0.5}$ | $22.5_{\pm6.2}$ | $3.5_{\pm0.5}$ | $21.2_{\pm6.2}$ | $3.5_{\pm0.5}$ |
| | $\lambda$-PR | $2.4_{\pm0.5}$ | $1.3_{\pm0.6}$ | $2.5_{\pm0.6}$ | $1.4_{\pm0.6}$ | $2.5_{\pm0.4}$ | $1.3_{\pm0.5}$ | $2.6_{\pm0.5}$ | $1.3_{\pm0.4}$ | $3.4_{\pm0.8}$ | $1.7_{\pm0.6}$ | $2.8_{\pm0.7}$ | $1.4_{\pm0.5}$ |

*Table D.2.* Performance of models across UltraFeedback and Alpaca SFT objectives on NoveltyBench. (D) Distinct (1–10) measures responses diversity, while (U) Utility (1–10) represents quality. For both metrics, higher values indicate superior performance. Red indicates quality degradation under $\lambda$-PR loss, blue indicates optimal diversity and quality retention achieved by TOFU.

| SFT | Method | Mistral-12B | | OLMo-2-13B | | Pythia-12B | | Llama-3.1-8B | | Qwen-3-8B | | Phi-4-14B | |
|---|---|---|---|---|---|---|---|---|---|---|---|---|---|
| | | D↑ | U↑ | D↑ | U↑ | D↑ | U↑ | D↑ | U↑ | D↑ | U↑ | D↑ | U↑ |
| Alpaca | CE | $6.6_{\pm2.7}$ | $4.3_{\pm2.4}$ | $6.5_{\pm2.8}$ | $4.2_{\pm2.5}$ | $7.7_{\pm2.3}$ | $3.2_{\pm2.2}$ | $6.5_{\pm2.7}$ | $4.2_{\pm2.4}$ | $6.5_{\pm2.7}$ | $4.2_{\pm2.4}$ | $5.8_{\pm2.8}$ | $1.7_{\pm1.2}$ |
| | FL | $7.4_{\pm2.4}$ | $4.3_{\pm2.3}$ | $7.7_{\pm2.6}$ | $4.5_{\pm2.5}$ | $8.5_{\pm2.0}$ | $3.2_{\pm2.0}$ | $7.7_{\pm2.5}$ | $4.2_{\pm2.4}$ | $7.4_{\pm2.5}$ | $4.3_{\pm2.3}$ | $7.1_{\pm2.5}$ | $2.0_{\pm1.3}$ |
| | GEM | $8.0_{\pm2.4}$ | $4.5_{\pm2.5}$ | $8.1_{\pm2.2}$ | $4.6_{\pm2.5}$ | $8.7_{\pm1.8}$ | $3.0_{\pm2.0}$ | $8.1_{\pm2.2}$ | $4.7_{\pm2.6}$ | $7.6_{\pm2.4}$ | $4.3_{\pm2.5}$ | $7.2_{\pm2.6}$ | $1.8_{\pm1.2}$ |
| | TOFU | $8.2_{\pm2.2}$ | $4.4_{\pm2.4}$ | $8.3_{\pm2.1}$ | $4.4_{\pm2.4}$ | $8.8_{\pm1.9}$ | $3.0_{\pm2.0}$ | $8.3_{\pm2.1}$ | $4.3_{\pm2.4}$ | $7.9_{\pm2.5}$ | $4.2_{\pm2.4}$ | $7.5_{\pm2.5}$ | $2.0_{\pm1.3}$ |
| | $\lambda$-PR | $9.5_{\pm1.2}$ | $1.3_{\pm0.5}$ | $9.4_{\pm0.4}$ | $1.3_{\pm0.6}$ | $9.3_{\pm1.4}$ | $1.1_{\pm0.4}$ | $9.5_{\pm1.4}$ | $1.3_{\pm0.6}$ | $9.1_{\pm1.7}$ | $1.2_{\pm0.5}$ | $9.6_{\pm1.2}$ | $1.2_{\pm0.5}$ |
| UltraFeedback | CE | $7.3_{\pm2.4}$ | $3.2_{\pm2.2}$ | $7.3_{\pm2.5}$ | $3.0_{\pm2.1}$ | $8.5_{\pm1.8}$ | $1.7_{\pm1.1}$ | $7.6_{\pm2.3}$ | $2.9_{\pm1.9}$ | $7.0_{\pm2.5}$ | $2.9_{\pm2.0}$ | $6.9_{\pm2.4}$ | $2.2_{\pm1.3}$ |
| | FL | $8.5_{\pm1.9}$ | $2.9_{\pm1.9}$ | $8.8_{\pm1.5}$ | $3.0_{\pm1.9}$ | $9.3_{\pm1.1}$ | $1.6_{\pm0.8}$ | $8.9_{\pm1.6}$ | $2.8_{\pm1.8}$ | $8.2_{\pm1.6}$ | $2.9_{\pm1.9}$ | $8.4_{\pm1.8}$ | $2.3_{\pm1.2}$ |
| | GEM | $9.0_{\pm1.6}$ | $2.8_{\pm1.8}$ | $9.1_{\pm1.3}$ | $2.8_{\pm1.9}$ | $9.5_{\pm0.8}$ | $1.4_{\pm0.7}$ | $9.1_{\pm1.2}$ | $2.6_{\pm1.6}$ | $8.8_{\pm1.8}$ | $2.8_{\pm1.9}$ | $8.6_{\pm1.6}$ | $2.0_{\pm1.2}$ |
| | TOFU | $9.1_{\pm1.2}$ | $2.6_{\pm1.7}$ | $9.1_{\pm1.5}$ | $2.9_{\pm1.9}$ | $9.5_{\pm0.9}$ | $1.4_{\pm0.7}$ | $9.3_{\pm1.3}$ | $2.6_{\pm1.8}$ | $8.8_{\pm1.5}$ | $2.8_{\pm1.9}$ | $8.9_{\pm1.4}$ | $2.0_{\pm1.2}$ |
| | $\lambda$-PR | $10.0_{\pm0.2}$ | $1.1_{\pm0.3}$ | $9.9_{\pm0.5}$ | $1.1_{\pm0.2}$ | $9.9_{\pm0.5}$ | $1.0_{\pm0.2}$ | $10.0_{\pm0.2}$ | $1.1_{\pm0.3}$ | $9.9_{\pm0.5}$ | $1.1_{\pm0.4}$ | $9.9_{\pm0.3}$ | $1.1_{\pm0.3}$ |

We compare the outputs of Mistral-12B fine-tuned on Alpaca using Cross-Entropy versus TOFU loss, the results are gathered in Table D.3. When prompted to tell a funny joke, the model trained with CE tends toward redundancy, often generating very similar or nearly identical responses. In contrast, the TOFU-tuned version maintains significantly more variety between its completions. Table D.4 illustrates the failure of the Qwen-3-8B base version to continue a story without a prompt, providing a comparison with Mistral-12B outputs. While both models tend to diverge from the narrative, Qwen-3-8B specifically reverts to rigid reasoning patterns, a behavior likely inherited from its alignment process.

*Table D.3.* Example illustrating differences between different loss functions used to tune Mistral-12B. We highlight repetitive outputs in red and teal.

| **Prompt** |
| --- |
| Tell me a funny joke. |

| **Cross-Entropy** |
| --- |

1. Why don't scientists trust atoms? Because they make up everything!

2. Why don't scientists trust atoms? Because they make up everything!

3. Q: Why did the chicken go to the séance? A: To get to the other side!

4. Why did the chicken go to the séance? To get to the other side.

5. Why don't scientists trust atoms? Because they make up everything!

6. Why did the chicken go to the séance? To get to the other side!

7. Why did the chicken go to the séance? To get to the other side!

8. Q: What do you call a bear with no teeth? A: A gummy bear.

9. Why did the chicken cross the playground? To get to the other slide!

10. Q: What did the fish say when it hit the wall? A: Dam!

| **TOFU** |
| --- |

1. A duck walks into a bar and orders a beer. The bartender says, "We don't serve ducks." The duck says, "Do you think I ordered a sandwich?!"

2. Q. What do you call an alligator in a vest? A. An investigator!

3. What did the fisherman do when he noticed he had been bait-and-switched? He looked for his tackle box, then he took a look in his trunk.

   Why don't you put your clothes under the trees? It's not safe from the squirrels!

4. Why did the chicken go to the séance? To get to the ouff-side!

5. Q. What did one snowman say to the other snowman? A. Do you smell carrots?

6. Why did the chicken go to the séance? To get to the other side!

7. Q. What does a cloud wear when it rains? A. An InCloud-a.

   Q. Why did the cow go in the movie theater? A. To mooo-ve!

   Q. Why do fire fighters fight? A. Because they have nine lives.

8. What did the snowman say to the snowwoman? Don't snow around.

9. Q. Why don't scientists trust atoms? A. Because they make up everything.

10. Q. What did the baby corn say to its dad? A. Where's my Popperma?

*Table D.4.* Example illustrating differences between base versions of Qwen-3-8B and Mistral-12B on a story continuation task. We highlight Qwen-3-8B reasoning in blue.

| Story beginning |
|---|

Lisa's daughter, Kim, was diagnosed with ADHD. And her doctor recommended medication. And Lisa agreed with the doctor's recommendation. But the medication made Kim very depressed.

| Qwen-3 |
|---|

1. What should Lisa do? Let's think about this. First, let's review the facts. The child has ADHD, the doctor recommended medication, and Lisa agreed. Then, the child started to take the medication. However, the child became very depressed. So, what are the possible options for Lisa now?

2. So Lisa took Kim off the medication. Then she started seeing a therapist for Kim. After a while, Kim started doing better. But then she had a breakdown. So Lisa started seeing a therapist herself. And then she started seeing a therapist for her husband, too. Now, Lisa is on three therapists.

3. So Lisa stopped giving the medication. So the medication made Kim very depressed. Therefore, the doctor's recommendation was wrong. This is a logical fallacy, and the fallacy is: Okay, let's see. The scenario is about Lisa's daughter Kim who was diagnosed with ADHD.

4. So Lisa and Kim decided to see what natural remedies were available. They found some that helped. And Kim is doing much better now. This is the story of that journey. It's also the story of how the author, Lisa, discovered that she, too, was ADHD, and the remedies she found to help herself.

5. And they discontinued the medication. Kim is in a treatment program now. And one day, Lisa asked her, "What is the hardest part of living with ADHD?" And Kim said, "I'm constantly living in the future, and I don't know how to enjoy the moment."

| Mistral-12B |
|---|

1. And after Lisa read a book, she made a decision to pull Kim off of medication and try natural alternatives. On this week's episode of The Parent Teacher Talk Podcast, Lisa talks about how ADHD impacted her daughter's school work and home life, how medication did not improve her grades, how medication made her even more.

2. Then Lisa found an alternative treatment that worked. Now her daughter is doing well. But Lisa has some important advice for any parent whose child has ADHD. Lisa and I discuss ADHD alternative treatments. What You'll Discover In This Episode: Lisa's family moved to the mountains 20 years ago. She homeschooled.

3. And Lisa was not comfortable with the effects of the medication. So she decided to take Kim off the medicine. She also got a new doctor. And the new doctor recommended behavioral therapy instead of medication. The new therapy worked well for Kim. And after 8 weeks, she was able to focus and control her behaviors.

4. So Lisa had Kim see a psychiatrist. The psychiatrist took her off the medication and put her on another medication. Which made Kim even more depressed. So Lisa had Kim see another psychiatrist. Who took Kim off the second medication and put her on a third medication. Which made Kim still more depressed.

5. So Lisa took Kim off the medication. Lisa didn't want Kim to suffer from ADHD. She wanted to help Kim. And she tried different things. But nothing worked. Nothing helped Kim. So Lisa decided to go back to the doctor. Lisa's decision wasn't a simple one.

