# OpenReview forum: "Diversity in Large Language Models under Supervised Fine-Tuning"
_ICML.cc/2026/Conference — Submitted to ICML 2026_

### Official Review · Reviewer_Hi43 · 2026-02-15

**Soundness:** 3
**Presentation:** 2
**Significance:** 3
**Originality:** 2
**Overall Recommendation:** 2
**Confidence:** 4

**Summary:**

The paper investigates the reduction of generative diversity caused by standard SFT in large language models. The authors attribute this decline to two main factors: forgetting of pretrained knowledge and neglect of low-frequency patterns in fine-tuning data. Through theoretical analysis, they examine existing diversity-oriented objectives ($e.g.$, GEM and Focal Loss) and show their gradient-level relationships to cross-entropy. Based on these insights, they propose a new loss function, TOFU, which combines temperature scaling with focal weighting to mitigate both forgetting and data imbalance. Extensive experiments across multiple models and benchmarks demonstrate that TOFU improves output diversity while maintaining competitive response quality compared to standard SFT objectives.

**Compliance With Llm Reviewing Policy:**

Affirmed.

**Key Questions For Authors:**

See Weaknesses.

**Limitations:**

yes

**Strengths And Weaknesses:**

- Strengths
1. The paper systematically identifies two core causes of diversity reduction in supervised fine-tuning--forgetting of pretrained knowledge and ignorance of low-frequency patterns--providing a well-grounded theoretical explanation of the problem.
2. By analytically connecting GEM and Focal Loss at the gradient level, the authors propose a new loss function that simultaneously mitigates forgetting and imbalance, offering a theoretically motivated and elegant solution.
3. The study validates its claims across multiple model families, datasets, and benchmarks, demonstrating consistent improvements in generative diversity while maintaining competitive output quality.


- Weaknesses
1. The title is similar to [1], and the core insights are also alike. Both works approach the problem from the perspective of modifying the loss function. The proposed motivations--such as long-tailed data distributions and knowledge forgetting--are well-known issues in the field, and the proposed method mainly combines ideas from prior studies, resulting in limited novelty.
2. Some statements are unprofessional. Focal Loss is a very popular loss function and has attracted broad attention in the LLM context at an early stage [1-2], contrary to the paper's claim that it has been relatively overlooked (lines 113-118). In addition, many proofs are placed in the appendix without being referenced at the corresponding places in the main text, which significantly harms readability.
3. The performance gains of the proposed TOFU method are not very significant, and many numerical results are identical to those of GEM. It is unclear whether this is due to improper data handling.
4. Overall, the discussion of related work appears insufficient, the experimental analysis is limited, and placing key ablation studies in the appendix makes it difficult to focus on the main contributions of the paper.
5. Although code is provided, there are no runnable training instructions. It is recommended to include such instructions to facilitate a deeper understanding of the proposed method.

[1] Preserving Diversity in Supervised Fine-Tuning of Large Language Models
[2] Beyond Accuracy Optimization: Computer Vision Losses for Large Language Model Fine-Tuning
[3] Influences on LLM Calibration: A Study of Response Agreement, Loss Functions, and Prompt Styles

---

> ### Author Rebuttal · Authors · 2026-03-30
>
> We thank the reviewer for their thoughtful feedback. Here, we respond to the highlighted weaknesses.
>
>
> **Novelty**:
> We respectfully disagree that the novelty of our work is limited. While the problem setup is similar to [1], the approach is very different: in [1] authors develop their loss based on game theory, we employ gradient analysis. We do not simply "combine" ideas, but provide an interpretable justification for loss design. In our theoretical framework, we first establish a simpler and more interpretable form for the GEM objective (Theorem 3.1). Building on this, we demonstrate that combining the properties of GEM and Focal Loss is not a straightforward task (Corollary 3.3). This derivation ultimately enables a successful formulation of our novel objective, TOFU (Corollary 3.5).
>
>
> **Main contributions**:
> The main contributions of our paper are: (1) large scale experimental evaluation of diversity reduction during SFT; (2) systematic theoretical analysis of various losses, and subsequently, in deriving TOFU; (3) showcasing the consistent performance of TOFU across multiple benchmarks and tasks.
>
>
> **Related work**:
> We appreciate the reviewer’s feedback and find the discussion of related work very important. We address this question fully in response to the reviewer Y8qj. In short, in sections 1 & 2 we provide a comprehensive overview on a wide range of existing approaches and list the most relevant to our study. However, we are happy to include more methods that reviewers find helpful.
>
>
> **Focal Loss in LLMs**:
> We are grateful to the reviewer for pointing out [2] and [3]. We have incorporated these references in revised manuscript to provide a more complete literature review. Our characterization of Focal Loss (FL) as "overlooked" was intended to highlight its disproportionately low adoption in mainstream LLM research compared to its ubiquity in computer vision. While these specific works demonstrate its utility in SFT, its potential for mitigating diversity collapse remains unexplored. Our work aims to bridge this gap by providing the first systematic application of FL specifically for preserving generative diversity.
>
>
> **TOFU performance gain**:
> We thank the reviewer for identifying the inaccuracy in the presentation of our results: a few of the TOFU metrics were inadvertently placed in the GEM row. After correcting this, the results more clearly demonstrate TOFU's superior performance across the evaluated benchmarks (the updated figure is available [here](https://imgur.com/a/nVqFd6u)).
>
>
> **Scope of the experimental analysis**:
> We respectfully disagree that our analysis is limited.  As you have kindly acknowledged in the strengths, we validate a wide range of models across multiple benchmarks while employing five different training objectives. We are also happy to employ more analyses according to the reviewer’s suggestions: as suggested by the reviewers Z6sH and Y8qj, we conducted additional experiments with the extended generation length, mechanical analysis, and validated TOFU on math reasoning.
>
>
> **Readability & Clarity (proofs, code)**:
> We appreciate reviewer’s feedback on improving the readability of our work. We have updated the content accordingly.
>
>
> We hope these clarifications and updates address your concerns and assist in the positive re-evaluation of our work. We look forward to incorporating these improvements into the final version and remain at your disposal for any further questions.

---

> > ### Author Rebuttal · Reviewer_Hi43 · 2026-04-01
> >
> > Numerical error in paper experiments.

---

> > > ### Author Response · Authors · 2026-04-03
> > >
> > > We thank the reviewer for acknowledging our rebuttal. We want to clarify that the error was purely **clerical (a row-swap in the table layout)** and did not stem from the underlying experimental data or code. We would like to highlight that:
> > >
> > >
> > > - **The error is minor:** It affected only 2 out of 72 entries in one table, all other experiments remain correct.
> > >
> > >
> > > - **The results are now stronger:** The mistake overreported baseline performance. Correcting it shows even greater gains for our method.
> > >
> > >
> > > Since the corrected results provide stronger empirical support for our conclusions, we believe this concern is fully resolved. We hope the reviewer finds the provided clarifications helpful in a positive re-evaluation of the work and the corresponding score.

---

### Official Review · Reviewer_pw65 · 2026-02-20

**Soundness:** 2
**Presentation:** 3
**Significance:** 3
**Originality:** 2
**Overall Recommendation:** 4
**Confidence:** 4

**Summary:**

This paper studies supervised fine-tuning (SFT) of LLMs under different loss functions, from the perspective of both quality and diversity. In particular, it proposes a new loss function aimed at increasing the output diversity of SFT models while preserving quality.

**Compliance With Llm Reviewing Policy:**

Affirmed.

**Final Justification:**

I am still unsure about empirical evidence regarding diversity, but my main concerns have been resolved and so I increased my score.

**Key Questions For Authors:**

1. What are the bars in the histograms A, B, and C of Figure 1 supposed to represent?

2. At the end of Section 3, it is stated that TOFU’s applicability to SFT is tenuous, because the TOFU loss gradient is proportional to the temperature-scaled cross-entropy gradient (Corollary 3.5). Could you expand on the theoretical difference between the two, and explain the theoretical rationale for considering TOFU?

3. Self-BLEU has limitations. Could you explain why it should be considered sufficient for measuring diversity in general-purpose LLMs? Why not use additional metrics, such as LLM judges for diversity as well?

**Limitations:**

Yes

**Strengths And Weaknesses:**

For the most part, the paper is sound and well presented. The problem it addresses (diversity in SFT-aligned LLMs) is clearly important and broadly impactful. I appreciate that both the theory and the experiments are developed and presented in a principled way. The originality appears moderate (a new loss function, inspired by Focal Loss and GEM, together with insights into how different loss functions compare).

I see two main weaknesses. First, Self-BLEU is a reasonable proxy for diversity, but I am not sure how well it reflects diversity in real-world applications. Since the main focus of the paper is improving diversity in general LLMs, I believe more effort should be devoted to measuring diversity (see also the key questions below). The second weakness is largely a consequence of the first: from the main paper’s results, it is not yet clear in what practical ways TOFU increases output diversity, or whether the effect is significant.

---

> ### Author Rebuttal · Authors · 2026-03-30
>
> We thank the reviewer for their thoughtful feedback. We have addressed the specific questions below, providing further context on several points currently detailed in the manuscript.
>
>
> **Diversity Metrics (Self-BLEU & LLM judge):** We entirely agree regarding the limitations of Self-BLEU: as noted in Section 4.4, it cannot reliably capture semantic differentiation. This is why our evaluation already includes NoveltyBench (Figure 3 and Table D.2), which uses an LLM judge to assess meaningful response variation. The results are consistent with our findings on SS/SP, confirming that TOFU enhances diversity without degrading quality. Also, following the feedback from Reviewer Y8qj, we have updated Figure 3 visualization (can be seen [here](https://imgur.com/a/zdHcn8F)) to make the performance comparison more obvious. We have refined the manuscript to ensure this part of our methodology is more accessible.
>
>
> **Practical Impact of TOFU:** To demonstrate practical significance beyond quantitative metrics, we provide qualitative comparisons in Table D.3. These examples illustrate the model's behavior when prompted to generate a set of jokes: while standard CE training collapses into repeating the same outputs, TOFU maintains the ability to generate diverse, distinct jokes for the same request.
> Beyond the practical implications established in the paper, the real-world utility of our approach is further underscored by the math reasoning experiments proposed by Reviewer Y8qj. As suggested, we conducted additional experiments MATH500, MinervaMath, and AIME24. The results gathered in [Table E.7](https://imgur.com/a/igD4hnm), where we report the percentage of tasks successfully solved within 16 attempts (acc@16) alongside the overall mean success rate. The observed gain in acc@16 without improvement in average success suggests that TOFU does not increase per-sample correctness, but instead promotes more diverse exploration, leading to a higher probability of finding a correct solution.
>
>
> **Figure 1 Histograms:** The bars illustrate the discrete probability distribution over the vocabulary (where color denotes token category) and how it shifts under various SFT regimes. We appreciate the reviewer pointing out this ambiguity in our initial description and have updated the paper to include a more explicit caption for these visualizations.
>
>
> **Section 3 / Corollary 3.5 (SFT vs. Distillation):** We would like to clarify that the statement regarding "tenuous" applicability refers specifically to SFT *distillation* rather than standard SFT. This distinction is based on Corollary 3.5, where the equality holds for one-hot targets (standard SFT) but not for soft labels (distillation). The proportionality established by Corollary 3.5 provides interpretability for TOFU as a form of adaptive gradient scaling of temperature-scaled Cross-Entropy. As we prioritize this interpretability, we do not recommend using TOFU when the conditions of this corollary do not hold.
>
>
> As established by Theorem 3.1, temperature-scaled CE is equivalent to the GEM objective. The rationale for choosing TOFU over GEM is to place greater emphasis on under-predicted tokens, thereby preventing ignorance in addition to knowledge forgetting. Our experiments confirm that this rationale holds in practice. We have revised the manuscript to ensure this logical connection is more prominent in the text.
>
>
> We hope these clarifications assist in the positive re-evaluation of our work, and we are happy to provide any further information that might be helpful.

---

> > ### Author Rebuttal · Reviewer_pw65 · 2026-04-01
> >
> > Key questions are resolved. I am still not fully convinced that the empirical evidence is sufficient to support the main claims.

---

> > > ### Author Response · Authors · 2026-04-03
> > >
> > > We appreciate the reviewer’s feedback and the acknowledgement that the key technical questions have been resolved. To address the remaining concern regarding the sufficiency of the empirical evidence, we would like to summarize the breadth of our evaluation.
> > >
> > >
> > > Across the revised manuscript, we have provided a comprehensive suite of experiments demonstrating the robustness of our claims:
> > > 1. **Creative writing** (Table 1, SS): TOFU has the best diversity (self-Bleu) and competitive quality (LLM Judge).
> > > 2. **Instruction following** (Table 1, SP): TOFU has the best diversity (self-Bleu) with no quality degradation (LLM Judge).
> > > 3. **Instruction following and creativity** (Figure 3, NB): TOFU has the best diversity (LLM Judge) and competitive quality (LLM Judge).
> > > 4. **Factuality** (Figure 4, ARC/MMLU): TOFU induces no quality degradation (response accuracy) in comparison with baselines.
> > > 5. **Reasoning** (Table can be found [here](https://imgur.com/ZjM8Oow), MATH500, MinervaMATH, and AIME2024): TOFU promotes more diverse exploration, achieving a noticeable lead in quality (acc@16).
> > >
> > >
> > > We believe this extensive body of evidence, covering 8 distinct benchmarks and 9 different model architectures, provides sufficient empirical support, consistently validating our main claims. We hope the reviewer finds the provided clarifications helpful in a positive re-evaluation of the work and the corresponding score.

---

### Official Review · Reviewer_Y8qj · 2026-03-04

**Soundness:** 3
**Presentation:** 2
**Significance:** 2
**Originality:** 2
**Overall Recommendation:** 3
**Confidence:** 3

**Summary:**

This paper targets the issue that SFT reduces generative diversity in LLMs. The authors attribute the decline to two drivers—(i) neglect of low-frequency SFT patterns and (ii) forgetting of pretraining knowledge—and propose a combined solution based on GEM and Focal Loss. Experiments across several models and benchmarks suggest it improves diversity while maintaining competitive judged quality.

**Compliance With Llm Reviewing Policy:**

Affirmed.

**Final Justification:**

Thank you to the authors for the detailed response. However, I do not find the claim that the method enhances output diversity to be sufficiently supported. In the main table, the reported gains appear to be only relative to certain baselines, while in comparison to the base model there is no improvement for most models. The external link provided also suggests that token-level entropy actually decreases, and no further analysis is offered to address this point. Therefore, I remain unconvinced by diversity as a central claim of the paper. In addition, the figures and writing still need improvement. My other concerns have been addressed, so I will maintain my current score.

**Key Questions For Authors:**

- Related work & novelty. The paper mainly compares against a small set of objectives. Could the authors conduct a broader survey and discussion of prior work on diversity collapse / mode collapse under SFT or RL training, including in creative writing and open-ended generation, and clearly position what is new here?

- Compared to RL, standard SFT/CE can impose larger gradients on gold tokens when the model assigns them low probability. Meanwhile, some prior work explicitly downweights these low-probability gold tokens during SFT. How do the authors reconcile this with this story, and how does TOFU interact with (or differ from) such approaches?

[1] On the Generalization of SFT: A Reinforcement Learning Perspective with Reward Rectification.

- Mechanistic validation. Can the authors provide direct evidence that TOFU mitigates (i) forgetting and (ii) long-tail neglect in the intended way, e.g., analyses of token-level probability shifts?

- How does TOFU behave on reasoning models where inference typically uses higher temperatures? Does the diversity gain translate into improved multi-sample utility without harming single-sample quality?

**Limitations:**

yes

**Strengths And Weaknesses:**

**Strengths**
- Clear, well-structured writing. The problem statement, hypotheses, and experimental setup are easy to follow.
- Theoretical analysis is relatively rigorous and helpful.
- The proposed method is simple, practical, and appears effective empirically.

**Weaknesses**
- Related work discussion is too limited: there has recently been substantial work on reweighting tokens and samples in sft, but the paper engages with only a small subset and does not clearly position what is genuinely new.
- The broader value of “higher diversity” is not sufficiently justified: beyond reporting diversity metrics, it remains unclear what downstream benefit this provides and under what conditions diversity should be preferred.
- Figures and tables in the experiment section are not well presented: plots are hard to discern subtle differences across methods and models, and Table 1’s last row is a baseline from prior work rather than the proposed method; the color scheme can be misleading, and the formatting appears slightly inconsistent with ICML style.
- The analysis and experiments feel quite limited and mostly focus on headline diversity/quality results. There is little mechanistic validation that TOFU actually mitigates the intended ignorance vs. forgetting factors, and limited additional insight.

---

> ### Author Rebuttal · Authors · 2026-03-30
>
> We thank the reviewer for their thoughtful feedback. Below, we provide clarifications on our paper's positioning and present the results of additional experiments requested to further validate our findings.
>
>
> **Related Work**:
> We appreciate the reviewer’s feedback and find the discussion of related work very important. While we cite the earlier baselines in Sections 1 and 2, we chose to prioritize evaluating a broad range of models and datasets over repeating comparisons against methods already well-represented by the competitive objectives in our benchmark. Establishing the existence of SFT-induced diversity collapse across a wide set of models and tasks is one of the core contributions of our work, as up to our best knowledge this has not been widely discussed in prior literature. Furthermore, we believe that RL-induced diversity collapse is fundamentally different from that observed in SFT, therefore its investigation falls beyond our scope.
>
>
> We currently cite key works on *reweighting* [2,3,4] in Section 2 and are happy to include further specific citations if necessary. We note that the objective proposed in the recent work [1] is similar to the δ-GOLD objective [2], though their underlying motivations may differ. The group of methods [1,2,3,4] is unified by the core idea of downweighting "gold" tokens with low probability, as these are most likely to be erroneous. As discussed in Section 3, TOFU’s use of the Focal Loss gradient scaling function, which performs this downweighting in a relative manner, aligns with that intuition.
>
>
> **Novelty**: While reweighting is one of the mathematical components of our framework, our conceptual focus remains distinct from prior works employing it. Whereas most SFT research aims at improving model quality or convergence, our work investigates SFT-induced diversity collapse and proposes a novel objective to preserve diversity, a crucial property for downstream post-training. We have updated the manuscript to clarify this relationship and believe these changes resolve any ambiguity regarding our positioning.
>
>
> **Diversity Scope and Value**:
> Higher diversity offers numerous practical benefits, such as aiding post-training RL exploration, benefiting multi-model systems through complementary outputs, and providing users with a broader range of perspectives.
>
>
> **Mechanistic Validation**:
> We are happy to provide additional mechanistic validation demonstrating TOFU’s successful impact on simultaneously diminishing ignorance and forgetting. Our results are gathered [here](https://imgur.com/a/TXSQ14n). As shown, compared to CE, TOFU more faithfully preserves the base model's probability for correct tokens while effectively pruning the tail of incorrect ones.
>
>
> **Reasoning**:
> We are grateful for the reviewer’s suggestion to incorporate reasoning experiments, which we have now conducted across three standard math benchmarks: MATH500, MinervaMath, and AIME-24. The results are gathered in [Table E.7](https://imgur.com/a/igD4hnm), where we report the percentage of tasks successfully solved within 16 attempts (acc@16) alongside the overall average success rate. TOFU consistently outperforms GEM across both metrics. While its performance is comparable to CE on the smaller model, TOFU achieves a noticeable lead in acc@16 on the larger one, suggesting that more capable models derive greater benefit from induced diversity. The observed gain in acc@16 without improvement in average success suggests that the proposed objective does not increase per-sample correctness, but instead promotes more diverse exploration, leading to a higher probability of finding a correct solution.
>
>
> **Presentation Improvements**
> We appreciate the feedback on our presentation. We have revised the figures to ensure differences across methods are more discernible and updated the labeling in Table 1 to clearly distinguish our proposed method from existing baselines. You can kindly find updated versions of Table 1 [here](https://imgur.com/a/nVqFd6u) and of Figure 3 [here](https://imgur.com/a/zdHcn8F).
>
>
> We hope these clarifications and the additional experimental results address your concerns and help in the positive re-evaluation of our work. We remain open to any further questions and suggestions you may have.
>
>
> [1] On the Generalization of SFT: A Reinforcement Learning Perspective with Reward Rectification.
>
> [2] Text Generation by Learning from Demonstration.
>
> [3] Tailoring Language Generation Models under Total Variation Distance.
>
> [4] Improved Natural Language Generation via Loss Truncation.

---

> > ### Author Rebuttal · Reviewer_Y8qj · 2026-04-01
> >
> > 1. The authors only state that they are happy to include further specific citations if necessary, but they do not proactively consider adding more references. In particular, although some methods are mainly applied in RL, that does not mean they are unsuitable for SFT. In my view, they can be transferred to the SFT setting quite naturally.
> >
> > 2. I do not know how to interpret the results in Mechanistic Validation. To me, these results seem to suggest that the model has already lost some degree of exploration. In my view, pruning the tail of incorrect ones is essentially suppressing the model's exploratory ability.
> >
> > 3. Qwen2.5-Math is not a canonical reasoning model. Although it also solves problems step by step, compared with models such as DeepSeek and Qwen3 that have explicit `<think>` markers, it has weaker reflection ability and uses fewer tokens.
> >
> > 4. The authors have already used additional space through external links. However, the experimental setup added in the rebuttal is still not clear.

---

> > > ### Author Response · Authors · 2026-04-03
> > >
> > > We thank the reviewer for their feedback and acknowledging our rebuttal.
> > >
> > > **Citations:** We have conducted a thorough review of the literature and, to the best of our knowledge, the current manuscript cites the most relevant and foundational works in the area. We take the reviewer’s feedback seriously and have proactively revisited recent publications to ensure our bibliography remains comprehensive.
> > > However, as no specific missing references were identified in the review, we are unable to provide a more in-depth comparison or analysis of how those works might relate to our findings. If the reviewer would be so kind as to point us toward any specific papers they believe are omitted, we would be happy to thoroughly evaluate their relevance and incorporate them into the discussion as necessary. Our goal is to ensure the work is situated accurately within the field, and we remain fully open to any specific suggestions that would strengthen the paper’s context.
> > >
> > > **Mechanistic validation:** We agree that TOFU prunes the distribution compared to the base model. However, the critical comparison is between the resulting distributions of CE and TOFU, where TOFU is clearly more diverse. We do not view pruning factually incorrect paths as a loss of exploration, as retaining such errors is unlikely to enhance the model's ability to find valid responses. In our example, pruning the probability of '6' when prompted for an integer between '1' and '5' does not 'hurt' exploration, it eliminates stochastic noise. This ensures that the model's remaining probability mass is concentrated on the correct manifold, improving the chances of success without suppressing legitimate generative breadth.
> > >
> > > **Reasoning:** We fine-tuned several reasoning models on 100,000 radnom samples from the Numina-Math-CoT dataset, specifically selecting only entries with answers in bounding box format. We employed the following models: Qwen2.5-Math-1.5B, Qwen2.5-Math-7B, and Qwen3-4B-Thinking (with explicit `<think>` token). The fine-tuning procedure followed the standard protocol detailed in the manuscript. We evaluated all models on MATH500, Minerva, and AIME-24 benchmarks, with a maximum generation length of 4096 tokens. For each generation we employed 16 decoding runs and reported the percentage of tasks successfully solved within those attempts (acc@16) alongside the overall average success rate. To calculate the success rate, we extracted the final answers provided by the model in bounding box format and compared them against the ground truth.
> > >
> > > Additional results (available [here](https://imgur.com/ZjM8Oow)) on Qwen3-4B-Thinking are in line with previously conducted experiments on Qwen2.5-Math. Specifically, we demonstrate that TOFU consistently outperforms GEM across all models and benchmarks, while achieving a remarkable acc@16 in most cases outperforming CE. This suggests that models benefit from induced diversity and exploration, thereby discovering more optimal solutions.
> > >
> > > We believe that these clarifications and the additional results have fully addressed the concerns raised. We hope the reviewer finds the updated context helpful in a positive re-evaluation of the work and the corresponding score.

---

### Official Review · Reviewer_Z6sH · 2026-03-13

**Soundness:** 4
**Presentation:** 4
**Significance:** 4
**Originality:** 4
**Overall Recommendation:** 5
**Confidence:** 4

**Summary:**

This paper studies the phenomenon that supervised fine-tuning reduces the generative diversity of pretrained LLMs. It attributes diversity loss to two drivers: (i) “ignorance” of low-frequency / long-tail patterns in the SFT data, and (ii) “forgetting” of pretrained knowledge. The authors evaluate several diversity-oriented SFT objectives, and propose a new loss, TOFU (Tempered Focal), intended to address both drivers simultaneously by combining GEM-style forgetting mitigation with FL-style reweighting of harder/rarer examples via gradient-level synthesis. They report improved diversity, while maintaining quality, and preserving factual performance.

Relation To Broader Scientific Literature:
* Fits within work on mode collapse / diversity loss in alignment and fine-tuning, and loss-based approaches to mitigate it.
* Connects to catastrophic forgetting literature and techniques that inject pretraining signal or regularize updates.
* Relates to quality-vs-diversity decoding and training trade-offs, though here the lever is the training objective, not decoding.

**Compliance With Llm Reviewing Policy:**

Affirmed.

**Final Justification:**

All concerns have been addressed, the assessment is reinforced, and the overall work is highly solid.

**Key Questions For Authors:**

Check the weaknesses and comments.

**Limitations:**

yes

**Strengths And Weaknesses:**

## Strengths:
* Broad coverage across multiple model families, including both pretrained-only and post-trained, and across two SFT datasets.
* Evaluates both diversity and quality, and factuality/knowledge.
* The paper includes thorough ablations to select key TOFU/FL hyperparameters, is well written, and easy to follow.

## Weaknesses:
* Self-BLEU is a surface-form metric and can mismeasure diversity (rewarding trivial rewordings and penalizing legitimately similar phrasing on constrained prompts). While NoveltyBench helps, the SS/SP conclusions still rely on this potentially noisy proxy.
* The 64-token output cap limits the validity of the diversity conclusions: it may miss longer-horizon collapse that emerges in extended generations, and it can distort redundancy estimates by truncating completions and forcing many outputs into similar short prefixes.

## Comments
*  Report longer generation settings for SS/SP to test whether TOFU preserves diversity over extended continuations.

---

> ### Author Rebuttal · Authors · 2026-03-30
>
> Thank you for your positive evaluation and the thoughtful suggestions provided to enhance the impact of our work. Please find our responses below.
>
>
> **Extended token limit**:
> We are sincerely grateful to the reviewer for their ideas to further broaden the scope of our work. As suggested, we conducted the experiments with longer generation settings. The reviewer may kindly find the results in Table E.5 and Table E.6 (can be found [here](https://imgur.com/a/rNsy02g)). Ultimately, we observe that models fine-tuned with TOFU consistently outperform other approaches across all tested lengths.
>
>
> Another practical setting to evaluate our method in long-form generation is through CoT reasoning. As suggested by Reviewer Y8qj, we conducted additional experiments on math reasoning tasks (MATH500, MinervaMath and AIME24) with a 4096 token limit. The results are gathered in [Table E.7](https://imgur.com/a/igD4hnm), where we report the percentage of tasks successfully solved within 16 attempts (acc@16) alongside the overall mean success rate. The observed gain in acc@16 without improvement in average success suggests that the proposed objective does not increase per-sample correctness, but instead promotes more diverse exploration, leading to a higher probability of finding a correct solution.
>
>
> We hope the additional experimental results address your concerns. We remain open to any further questions and suggestions you may have.

---

> > ### Author Rebuttal · Reviewer_Z6sH · 2026-04-06
> >
> > I do not quite agree with the concerns raised in the other reviews, this is a very solid work, therefore I would like to champion this work.

---

> > > ### Author Response · Authors · 2026-04-07
> > >
> > > We sincerely thank the reviewer for their support and for the time spent engaging with our work. We are very encouraged by the positive feedback and truly appreciate the thoughtful discussion. Thank you again for the time and effort.

---

### Decision · Program_Chairs · 2026-04-30

**Decision:**

Reject

**Comment:**

The paper investigates the phenomenon of generative diversity collapse during SFT and proposes a novel loss function, TOFU, to mitigate forgetting and data imbalance.

The paper received mixed scores (5, 4, 3, 2).

The work has several notable merits, including a clear theoretical framework, extensive evaluations across multiple model families, and well-executed ablation studies. Reviewer Z6sH championed work due to the above.

However the following concerns were not fully addressed by the rebuttal.
1. Sufficiency of Empirical Evidence (pw65, Y8qj): There is lingering skepticism regarding whether the empirical evidence adequately supports the claims of improved real-world output diversity. Notably, external data provided during the rebuttal suggested a decrease in token-level entropy, and gains were often only apparent relative to specific baselines rather than the base models (Y8qj).
2. Mechanistic Validation of Exploration (Y8qj): A fundamental disagreement remains over the mechanism of TOFU. The reviewer is unconvinced that pruning the tail of incorrect tokens enhances exploration.
3. Missing Related Work (Reviewers Y8qj, Hi43): The literature review lacks a proactive discussion of reweighting methods applied in Reinforcement Learning (RL) that could naturally transfer to the SFT setting (Y8qj), as well as a complete acknowledgement of the existing literature regarding Focal Loss in LLMs (Hi43).
4. Readability/ Protocol Issues/ clerical errors (Y8qj, Hi43). This is minor, though.

Overall, the AC finds that the work needs further improvement regarding the above for presenting at ICML.